# The AKT2/SIRT5/TFEB pathway as a potential therapeutic target in non-neovascular AMD

Sayan Ghosh[1], Ruchi Sharma[2], Sridhar Bammidi[1], Victoria Koontz[1], Mihir Nemani[1], Meysam Yazdankhah[1], Katarzyna M. Kedziora[3], Donna Beer Stolz[3], Callen T. Wallace [3], Cheng Yu-Wei [4], Jonathan Franks[3], Devika Bose[2], Peng Shang[5], Helena M. Ambrosino[5], James R. Dutton[6], Zhaohui Geng[6], Jair Montford[2], Jiwon Ryu [2], Dhivyaa Rajasundaram[7], Stacey Hose[1], José-Alain Sahel [1,8], Rosa Puertollano[9], Toren Finkel [4], J. Samuel Zigler Jr[10], Yuri Sergeev [11], Simon C. Watkins [3], Eric S. Goetzman[7], Deborah A. Ferrington[5,12], Miguel Flores-Bellver [13], Kai Kaarniranta[14,15], Akrit Sodhi [10], Kapil Bharti[2] ✉, James T. Handa [10] ✉ & Debasish Sinha [1,10] ✉

Non-neovascular or dry age-related macular degeneration (AMD) is a multifactorial disease with degeneration of the aging retinal-pigmented epithelium (RPE). Lysosomes play a crucial role in RPE health via phagocytosis and autophagy, which are regulated by transcription factor EB/E3 (TFEB/E3). Here, we find that increased AKT2 inhibits PGC-1α to downregulate SIRT5, which we identify as an AKT2 binding partner. Crosstalk between SIRT5 and AKT2 facilitates TFEB-dependent lysosomal function in the RPE. AKT2/SIRT5/TFEB pathway inhibition in the RPE induced lysosome/autophagy signaling abnormalities, disrupted mitochondrial function and induced release of debris contributing to drusen. Accordingly, AKT2 overexpression in the RPE caused a dry AMD-like phenotype in aging *Akt2* KI mice, as evident from decline in retinal function. Importantly, we show that induced pluripotent stem cell-derived RPE encoding the major risk variant associated with AMD (complement factor H; CFH Y402H) express increased AKT2, impairing TFEB/TFE3-dependent lysosomal function. Collectively, these findings suggest that targeting the AKT2/SIRT5/TFEB pathway may be an effective therapy to delay the progression of dry AMD.

Non-neovascular or dry AMD is a leading cause of blindness among the elderly population worldwide[1,2]. Available treatments slow but do not prevent its progression[3], highlighting the importance of identifying and developing treatment targets for this blinding disease. The homeostasis of the neural retina is dependent on the underlying retinal pigmented epithelium (RPE)[4]. Specifically, to maintain photoreceptor viability needed for vision, the RPE phagocytoses daily, photoreceptor outer segments, a process that relies heavily on

lysosomal-mediated waste clearance/autophagy[1,4–6]. The RPE provides multiple other sight-saving functions that require a high metabolic rate that is driven by its abundant mitochondria, which also produce substantial and potentially toxic waste[4,5]. Since the RPE are post-mitotic, they also rely heavily on lysosomal-mediated clearance for their own health and survival[1,5,6]. Disruption of lysosomal and autophagy processes induces RPE dysfunction and visual disturbance. In early AMD, the RPE has impaired autophagy and is

among the first cells that become dysfunctional and contribute mechanistically to AMD pathobiology[1,5,6].

Lysosomal biogenesis and function are under the control of the master regulator, transcription factor EB and/or E3 (TFEB and TFE3)[7]. TFEB and TFE3 belong to the MiTF/TFE (basic helix-loop-helix) transcription factor subfamily that regulates the transcription of genes in the lysosome-autophagy pathway and is, in turn, regulated by the mechanistic target of rapamycin, complex 1 (mTORC1)[7]. TFEB/TFE3 can also be regulated by mTOR-independent pathways such as AKT1/2 and calcium signaling[8–10]. AKT has been reported to phosphorylate TFEB at Ser467 to repress TFEB nuclear translocation independently of mTORC1[8]. AKT2 is both an upstream regulator of mTORC1 and a downstream mediator for mTORC2[11,12], and thus, plays a central role in regulating lysosomal function and autophagy. While increased AKT2 has been observed in the retina and the macular RPE of human AMD donors[13,14], the underlying role of AKT2-dependent signaling in AMD progression remains elusive.

AMD is a multi-factorial disease where age, smoking, and genetic variants are among the highest risk factors[1,15–19]. Genome Wide Association Studies (GWAS) have identified genetic variants in the complement system, extracellular matrix remodeling, lipid metabolism, and lysosomal biogenesis that are associated with increased risk for AMD[15–17]. In particular, the CFH-CFHR5 region on chromosome 1q32 is associated with very strong AMD risk[18,19]. Interestingly, alterations in lysosomal function and autophagy have been reported in iPSC-derived cells from donors with *CFH* Y402H[20]. However, the impact of this AMD-associated genetic risk allele and its downstream effects, and how these alterations contribute to AMD pathogenesis are unknown. While a role for AKT1 and AKT3 in lysosomal function cannot be excluded, here we show that AKT2 plays an important role in regulating lysosomal and mitochondrial function in RPE cells. Given the central role of AKT2 in maintaining these cellular processes, the present study was undertaken to establish the role of altered AKT2-dependent signaling on lysosomal function and autophagy in the RPE that contributes to AMD pathology and the impact of AKT2 signaling in the RPE on the AMD phenotype and the influence of the CFH Y402H variant on lysosomal function.

## Results

### AKT2 upregulation in the RPE triggers lysosomal dysfunction and a dry AMD-like phenotype

It was previously reported that AKT2 is upregulated in dysmorphic macular RPE cells from AMD donor globes[13,14]. Moreover, in a well-characterized mouse model of lysosomal impairment that develops an AMD-like phenotype (the *Cryba1* cKO), we previously found that AKT2 is increased in the RPE, which caused impaired lysosomal biogenesis[21,22]. However, whether dysfunctional AKT2 signaling directly contributes to AMD progression is not known. To explore this question, we generated an RPE-specific *Akt2* knockin (KI) mouse, as previously described[23], and verified that the AKT2 upregulation is specific to the RPE and not present in the neurosensory retina (Supplementary Fig. 1a). Furthermore, AKT1 levels are unchanged[23]. The *Akt2* KI mice developed an AMD-like phenotype, beginning at 10 months, and had mild phenotypic worsening by 15 months old. For example, 10-month-old *Akt2* KI mice accumulated autofluorescent foci compared to age-matched controls, as seen on fundus images (Supplementary Fig. 1b). The RPE of 10-month-old *Akt2* KI mice were morphologically heterogeneous with a significantly higher number of dysmorphic cells intermixed among the normal cobblestone shaped cells in both the peripheral and central retina, while WT age-matched controls had uniform RPE morphology (Fig. 1a and Supplementary Fig. 1c). By 12 months, the dysmorphic RPE of *Akt2* KI mice had reduced apical microvilli length relative to age-matched WT mice (Fig. 1b). We also found decreased immunolabeling of

Phosphoprotein 50 (EBP50), which binds to the apically oriented Ezrin-Radixin-Moesin complex, which is critical for maintaining RPE polarity[24] (Fig. 1b). Ezrin itself was also significantly decreased in the RPE of *Akt2* KI mice (Supplementary Fig. 1e). Neural retinal degeneration also developed coincident with the RPE alterations. In 12-month-old *Akt2* KI mice, rhodopsin was diffusely stained, as seen with immunofluorescence studies, and had significantly reduced expression (Fig. 1b). This photoreceptor loss correlated with outer nuclear layer (ONL) thickness thinning beginning as early as in 10-month-old *Akt2* KI retinas (Supplementary Fig. 1g). As the mice aged to 15 months old, RPE apical microvilli had disintegrated and circular shaped photoreceptor outer segments, suggestive of degeneration, were observed by transmission electron microscopy (TEM); these changes were not observed in young mice (Supplementary Fig. 1d). In 15-month-old *Akt2* KI mice, the RPE had accumulated lipid deposits that were visualized with Perilipin-2 (PLIN2) staining[21] (Supplementary Fig. 1f). The RPE had truncated and fewer basal infoldings adjacent to Bruch's membrane basal laminar deposits (BLamD) in 15-month-old *Akt2* KI mice, which were not seen in age-matched controls (Fig. 1c). These histologic and corresponding molecular changes correlated with decreased a- and b-wave amplitudes on electroretinograms (ERG) of old *Akt2* KI mice while no ERG changes were observed in age-matched WT control mice (Supplementary Fig. 1h, i). Collectively, these morphologic, ultrastructural, molecular, and physiologic studies suggest that activated AKT2 signaling directly contributes to an early, dry AMD phenotype.

To assess the contribution of lysosomal dysfunction to the AMD phenotype in the RPE of *Akt2* KI mice, we examined the expression of a group of genes in these cells, known as the Coordinated Lysosomal Expression and Regulation (CLEAR) network genes, which are associated with lysosomal biogenesis acidification and the autophagy pathway[7]. We observed that CLEAR network genes were downregulated in the RPE of *Akt2* KI mice compared to wild-type littermate controls (Fig. 1d). Conversely, no changes in CLEAR gene expression were observed in *Akt2* conditional knockout (cKO) RPE (Fig. 1e). Moreover, we found that TFEB and TFE3 phosphorylation was increased in the RPE of *Akt2* KI mice relative to WT (Supplementary Fig. 2a). In TEM micrographs, we found a significant increase in the cumulative lysosomal area (normalized to the total tissue area) in *Akt2* KI RPE cells, compared to WT (Supplementary Fig. 2b), indicating alterations in lysosomal biogenesis[20]. AKT is known to regulate TFEB via an mTOR-independent pathway[8]. It has previously been reported that in cells with diminished TFEB nuclear activity lysosomal function can be compensated by TFE3 activation[25]. We found that upregulating AKT2 in TFEB null (*Tfeb* KO) mouse embryonic fibroblasts (MEFs) increased TFE3 phosphorylation (S321) and reduced expression of CLEAR genes, including Cathepsin D (CTSD) and L (CTSL), major lysosomal enzymes required for photoreceptor outer segment degradation[5,20] (Supplementary Fig. 3a, b). In addition, AKT2 overexpression in *Tfeb* KO cells induced TFE3 binding to 14-3-3 proteins, which is known to sequester TFE3 in the cytoplasm[7] (Supplementary Fig. 3c). These results indicate that TFE3 can compensate for TFEB loss and mitigate lysosomal dysfunction.

### AKT2 upregulation in RPE cells is associated with loss of lysosomal function in the RPE from human AMD donors and iPSC-derived RPE with the CFH Y402H risk allele

In several landmark papers, GWAS was used to identify over 50 single nucleotide polymorphisms (SNPs) that confer AMD risk[26,27]. Many of these SNPs affect genes involved in the complement cascade[18,19]. Furthermore, complement components have been identified in drusen[28]. Specifically, the complement factor H (CFH)-CFH Receptor 5 (CFHR5) region on chromosome 1q32 is a major genetic risk locus for AMD, increasing the odds of disease by more than 2.5-fold among heterozygotes and 7.5-fold among homozygotes[18,19]. iPSC-derived RPE cells

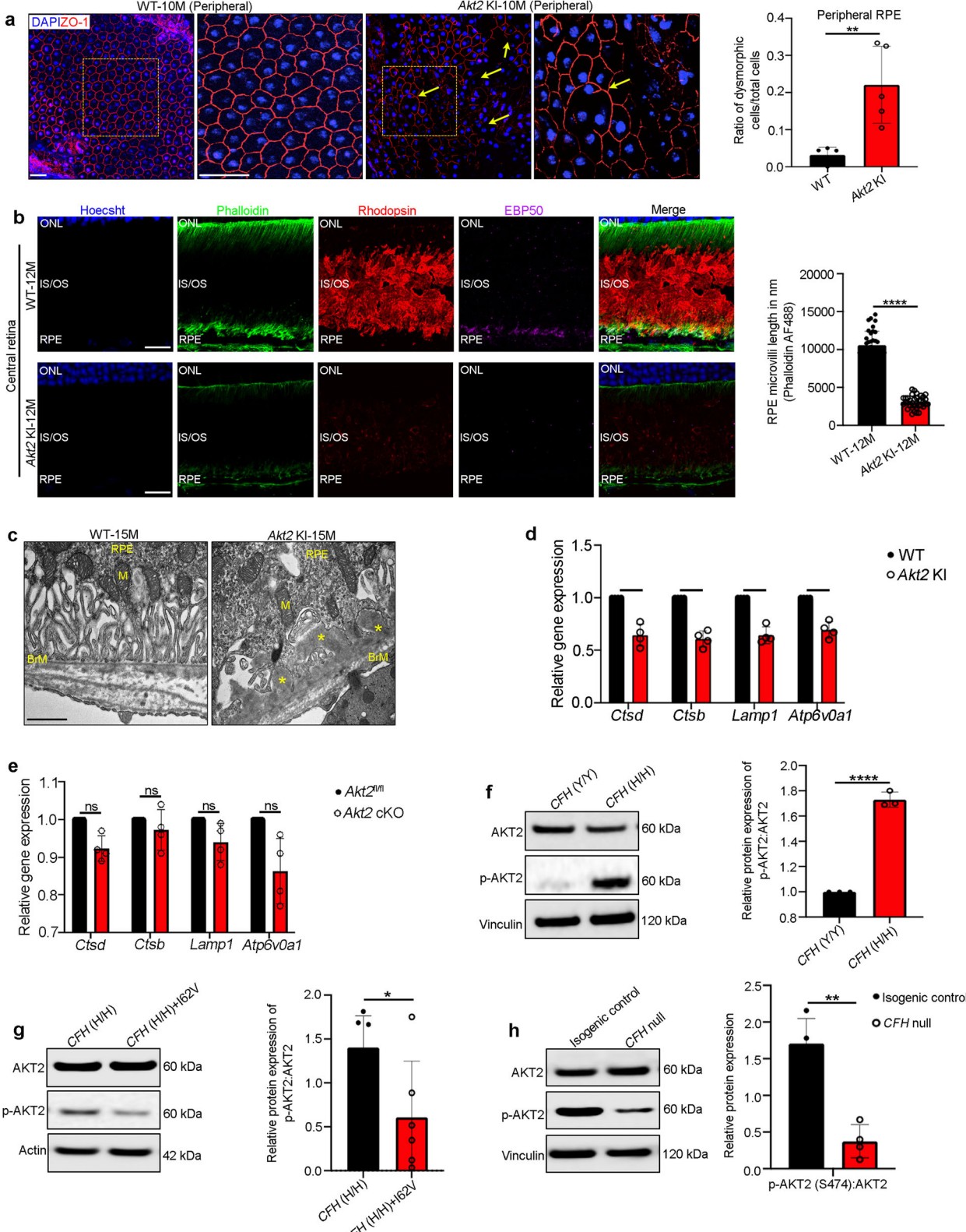

from donors with the *CFH* 402H risk allele have been reported to demonstrate defects in lysosomal function[20]. However, a molecular mechanism explaining how *CFH* Y402H contributes to AMD risk has remained elusive. We, therefore, set out to determine whether the *CFH* risk alleles influence the regulation of lysosomal function by AKT2 signaling. To this end, we generated RPE cells from iPSCs isolated from non-AMD human donors that were either homozygous for the

*CFH* wild type [402Y (Y/Y)] or the risk allele [402H (H/H)][29]. The quality of the iPSC-derived RPE cells was evaluated by transepithelial resistance (TER), expression of RPE markers like melanosome marker PMEL17 and Ezrin along with morphologic assessment using ZO-1 staining, and melanin pigmentation, as previously described[29]. All the cell lines used in this study showed robust TER values, cobblestone RPE morphology with pigmentation, and PMEL17 and Ezrin expression

**Fig. 1 | Akt2 upregulation triggers lysosomal dysfunction and an AMD-like phenotype in mice. a** Quantification of ZO-1 immunostaining on RPE flatmounts from 10-month-old *Akt2* KI mice shows an increased number of dysmorphic RPE cells in the peripheral regions (arrows), not seen in age-matched WT RPE. n (biological replicate)=5. Scale bar = 50 μm. **b** Immunofluorescence studies revealed a noticeable decrease in EBP50 (magenta) and rhodopsin (red) staining in the RPE, as well as decreased microvilli length (green and graph) in retinal sections, from 12-month-old *Akt2* KI mice, compared to WT. n (biological replicate) = 4. Datapoints show individual microvili lengths at different regions in the retina sections of 4 different biological replicates. Scale bar = 50 μm. **c** Transmission electron micrographs showing accumulation of basal laminar deposits in the RPE cells (asterisks) above Bruch's membrane (BrM) in 15-month-old *Akt2* KI mice, but not in age-matched WT RPE cells. M =mitochondria. n = 5. Scale bar 600 nm. **d**, **e** Statistically significant decline in expression of CLEAR network genes *Ctsd*, *Ctsb*, *Lamp1*, and

*Atp6v0a1* in *Akt2* KI, but not in *Akt2* cKO RPE cells. n (biological replicate)=4. **f** Western blot analysis showing elevated p-Akt2:Akt2 ratio in iPSC-derived RPE cells from *CFH* Y402H risk allele [homozygous; *CFH* (H/H)] containing donors (with no AMD), relative to controls [*CFH* (Y/Y)] (n; biological replicate = 3), and **g** reduction of the p-Akt2:Akt2 ratio when *CFH* I62V mutation is also present in CFH (H/H) cells (n; biological replicate = 6) or **h** by complete knockout of *CFH* (*CFH* null) in iPSC-derived RPE cells. n (biological replicate) = 3. All values are Mean ± S.D. *P < 0.05, **P < 0.01, ****P < 0.0001. The statistical test used in (**a**, **b**, **d**–**h**) is Student's t-test (Two-sided). The exact p-values are **a** P = 0.040 (*Akt2* KI vs WT); **b** P = 2.201E-22 (*Akt2* KI vs WT); **d** *Ctsd*: P = 0.0257, *Ctsb*: P = 0.0079, *Lamp1*: 0.0118, *Atp6v0a1*: P = 0.0177 (*Akt2* KI vs WT); **e** *Ctsd*: P = 0.0770, *Ctsb*: P = 0.8538, *Lamp1*: 0.3277, *Atp6v0a1*: P = 0.1918 (*Akt2* cKO vs *Akt2^fl/fl*). **f** P = 0.000092 (*CFH* (H/H) vs *CFH* (Y/Y)); **g** P = 0.0469 (*CFH* (H/H) + I62V vs *CFH* (H/H)); **h** P = 0.0052 (*Cfh* null vs Isogenic control). Source Data is provided in the Source Data file.

(Supplementary Fig. 4 and Supplementary Table 1). AKT2 phosphorylation was upregulated in *CFH* (H/H) cells compared to *CFH* (Y/Y) iPSC-derived RPE cells[29] (Fig. 1f). The *CFH* I62V allele[18] is protective for AMD risk. To assess whether the protection provided by the *CFH* I62V allele could also be mediated through AKT2, we observed a reduction in AKT2 phosphorylation in iPSC-derived RPE cells that expressed both *CFH* I62V in *CFH* (H/H) (Fig. 1g). Interestingly, *CFH* null (*CFH^{-/-}*) iRPE (induced RPE) cells also demonstrated decreased AKT2 phosphorylation (Fig. 1h).

### Akt2 upregulation in the RPE cells is associated with loss of lysosomal function in AMD

To explore whether the *CFH* risk allele is linked to elevated levels of AKT2 and deficiencies of lysosomal function, we examined lysosomal proteins in RPE cells with the CFH 402H risk allele. We observed that the increased levels of AKT2 and phosphorylated TFEB and TFE3 in *CFH* (H/H) cells, correlated with decreased expression and activity of the lysosomal proteins Cathepsin D and L (Fig. 2a–c), as has been previously reported[20]. Notably, this decline in lysosomal function was not observed in human *CFH* null iRPE cells (Supplementary Fig. 5a, b), raising the possibility that the presence of the 402H variant activates downstream signaling pathways[29], and in particular the AKT2-mediated lysosomal abnormalities. We also observed increased phosphorylation of Akt2 and both TFE3 and TFEB in RPE lysates from human AMD donors; this observation was not seen in age-matched non-AMD controls (Fig. 2d). While an increase in total phosphorylation of these transcription factors does not necessarily reflect a decrease in their nuclear translocation, it has previously been reported that serine phosphorylation at specific residues in TFEB (S211) and TFE3 (S321) is associated with reduced nuclear translocation of these proteins in multiple cell types[7–10,25]. In accordance with these findings, we examined AMD patient eyes and observed that they tend to show decreased nuclear TFEB immunostaining compared to non-AMD control eyes (Supplementary Fig. 6a–d). Finally, we observed that Cathepsin D and L protein levels and activities were also decreased in human AMD donor RPE, compared to controls (Fig. 2e–g). Collectively, these observations suggest that AKT2-mediated lysosomal dysfunction explains in part, the contribution of *CFH* risk alleles to AMD pathogenesis.

### AKT2/SIRT5 signaling axis regulates TFEB-dependent lysosomal function

We next set out to identify the mechanism by which increased AKT2 controls the nuclear translocation of TFEB/TFE3 and thereby lysosomal biogenesis. To this end, we performed a high-throughput human protein–protein array[22] and identified Sirtuin 5 (SIRT5) as the binding partner with the highest Z-score for AKT2 (Supplementary Table 2). This finding was particularly notable because sirtuins have previously been implicated in several metabolic and age-related diseases[30]. In particular, SIRT5 regulates both mitochondrial function and

autophagy[31]. SIRT5 has also been shown to play a critical role in aging and has been identified as a possible therapeutic target for age-related diseases[30]. Using computer modeling, we found that AKT2 (orange) may bind directly to SIRT5 (blue) (Fig. 3a). Indeed, co-immunoprecipitation studies using ARPE19 cells overexpressing SIRT5-HA-pcDNA and AKT2-GFP-pcDNA demonstrated that SIRT5 does bind to AKT2 (Fig. 3b). To determine if SIRT5 can influence phosphorylation of TFEB by AKT2, we incubated anti-GFP pulldown complexes from ARPE19 cells overexpressing AKT2-GFP or AKT2-GFP+SIRT5-HA with recombinant human TFEB and found that TFEB phosphorylation was reduced in AKT2 pulldown complexes from cells that overexpressed SIRT5 compared to cells overexpressing only AKT2 (Fig. 3c). In addition, treatment with the known AKT2 inhibitor (CCT128930)[22] at a dose of 10 nM for 30 min at 37 °C to the AKT2 pull-down complex also diminished TFEB phosphorylation (Fig. 3c), indicating that AKT2 is critical for regulating TFEB phosphorylation. SIRT5 carries out several post-translational modifications like malonylation, succinylation, acetylation, and glutarylation that regulate several signaling pathways both in the cytosol and mitochondria[30,31]. Since SIRT5 and AKT2 both localize to cytosol and mitochondria[8,22,23,30,31], we therefore wanted to ascertain if SIRT5 can regulate AKT2 signaling and, in particular, lysosomal function. We performed western blots to assess lysosomal mediators in ARPE19 cells overexpressing AKT2-GFP and SIRT5-HA+AKT2-GFP constructs. Our results showed that co-transfection with both AKT2 and SIRT5 constructs could reduce p-TFEB (S211) levels (phosphorylation site responsible for diminished nuclear translocation of this transcription factor), compared to only AKT2 overexpressing cells (Fig. 3d), suggesting that SIRT5 may regulate AKT2-mediated lysosomal function.

AKT2 has been reported to inhibit peroxisome proliferator-activated receptor-coactivator 1 alpha (PGC-1α)[32], which can upregulate SIRT5[33]. We therefore postulated that PGC-1α may facilitate crosstalk between SIRT5 and AKT2. Specifically, we reasoned that in the RPE of *Akt2* KI mice, SIRT5 abundance may be mediated by the AKT2/PGC-1α axis. We observed that both PGC-1α and SIRT5 expression were decreased in the RPE of *Akt2* KI mice compared to WT mice (Fig. 3e). Conversely, *Akt2* cKO RPE cells demonstrated an increase in both proteins relative to floxed controls (Supplementary Fig. 7). Furthermore, SIRT5 was decreased and p-AKT2 (Ser474) was increased in RPE lysates from both *PGC-1α^{-/-}* and *Sirt5^{-/-}* mice[34,35] (Fig. 3f, g). Importantly, in *CFH* (H/H) iPSC-RPE cells, which have high AKT2 signaling, both SIRT5 and PGC-1α were decreased as compared to control *CFH* (Y/Y) cells (Fig. 3h). Collectively, these results suggest that AKT2 regulates SIRT5 expression through PGC-1α, whereas SIRT5 can regulate AKT2 activity and subsequently phosphorylation of AKT2 targets, including TFEB.

Since PGC-1α can regulate mitochondrial biogenesis[33,34], we next explored mitochondrial function in ARPE19 cells overexpressing either AKT2 or AKT2+SIRT5 by measuring the oxygen consumption rate (OCR) and extracellular acidification rate (ECAR) using the Seahorse

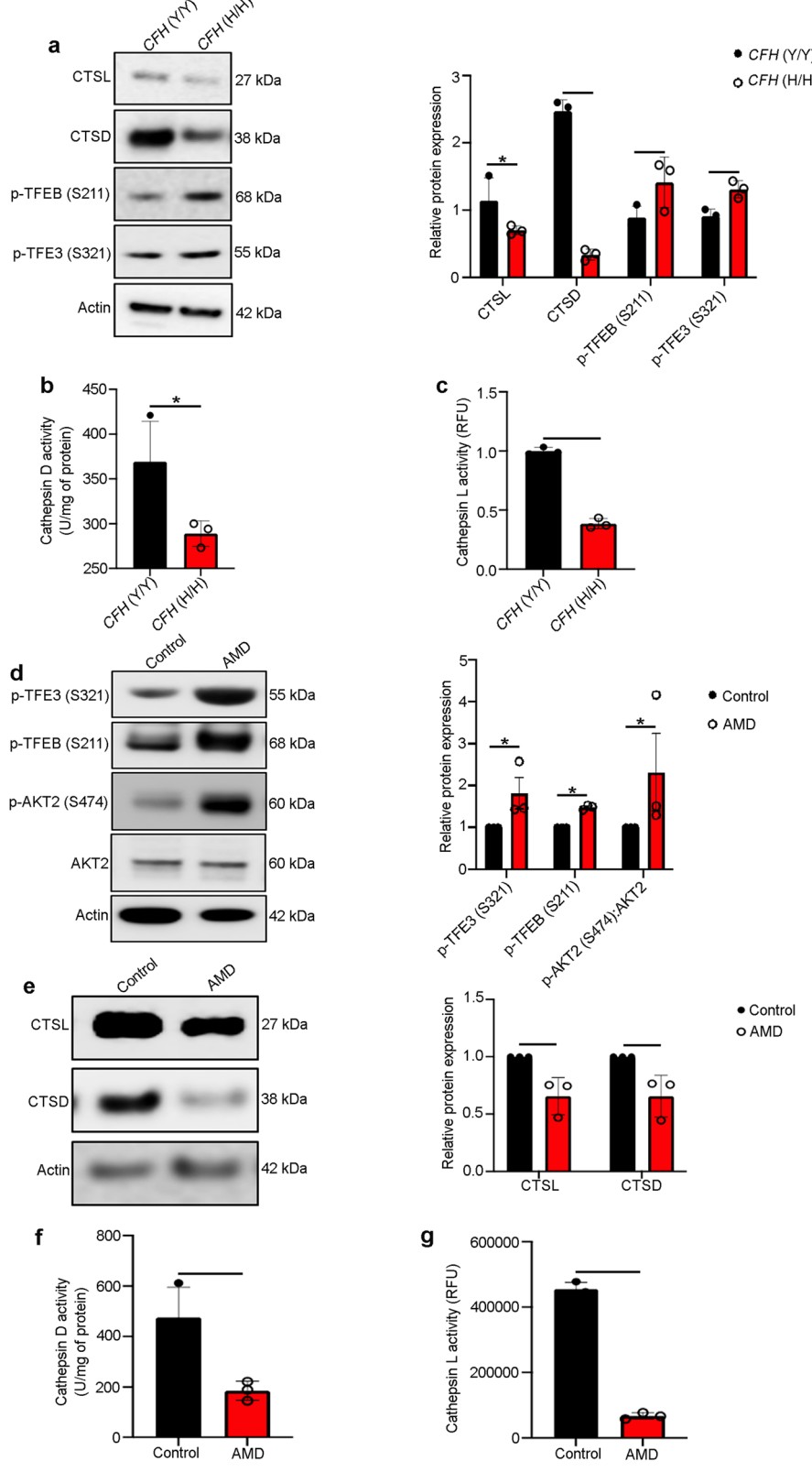

assay[36]. While both maximal respiration and ATP-linked respiration were impaired in AKT2 overexpressing cells, they were rescued by the addition of SIRT5 overexpression (Supplementary Fig. 8a–c). These data suggest that a functional connection between lysosomes and mitochondria is essential for maintaining RPE homeostasis. While dysfunction of either organelle has been shown to play a pivotal role in AMD pathobiology, these results suggest that abnormalities in

lysosome-mitochondria crosstalk could also adversely impact RPE health and contribute to AMD pathobiology.

## AKT2 upregulation in the RPE cells deregulates macro-autophagy and induces secretory autophagy

Lysosomal dysfunction contributes to RPE degeneration and the formation of extracellular deposits in Bruch's membrane called drusen,

**Fig. 2 | Akt2 upregulation in the RPE cells is associated with loss of lysosomal function in AMD. a** Western blot showing elevated p-TFEB (S211) and p-TFE3 (S321) levels along with decreased levels of lysosomal hydrolases CTSD and CTSL in CFH(H/H) cells relative to CFH (Y/Y) cells. n (biological replicate) = 3.
**b, c** Colorimetric analysis revealed significant downregulation of activities of both CTSD and CTSL in iPSC-derived RPE cells from *CFH* Y402H risk allele [*CFH* (H/H)] containing donors, relative to controls. n (biological replicate) = 3. **d** Western blot analysis showing elevated levels of p-Akt2 (S474), p-TFE3 (S321), and p-TFEB (S211) and **e** downregulation of CTSD and CTSL in RPE lysates from human AMD donors, compared to age-matched controls. n (biological replicate) = 3. **f, g** RPE lysates from

human AMD donors also showed significant downregulation of both CTSD and CTSL activities, compared to controls. n (biological replicate) = 3. All values are Mean ± S.D. ****$P < 0.0001$, ***$P < 0.001$, **$P < 0.01$, *$P < 0.05$. The statistical test used in (**a**–**g**) is Student's t-test (Two-sided). The exact p-values are **a** CTSL: P = 0.0422, CTSD: P = 0.00028274, p-TFEB (S211): P = 0.0498, p-TFE3 (S321): P = 0.0142 (*CFH* (H/H) vs *CFH* (Y/Y)); **b** P = 0.0419 (*CFH* (H/H) vs *CFH* (Y/Y)); **c** P = 8.585E-05 (*CFH* (H/H) vs *CFH* (Y/Y)); **d** p-TFE3 (S321): P = 0.0488, p-TFEB (S211): P = 0.0041, p-AKT2 (S474): AKT2: P = 0.0491 (AMD vs Control); **e** CTSL: P = 0.0175, CTSD: P = 0.0173 (AMD vs Control); **f** P = 0.042 (AMD vs Control); **g** P = 7.824E-05 (AMD vs Control). Source Data is provided in the Source Data file.

two hallmarks of early AMD[1,5,6]. Despite intensive research, the pathogenesis of drusen remains unclear. We set out to determine if malfunction of lysosome-mediated physiological processes, such as autophagy, could promote the extracellular waste accumulation that contributes to drusen biogenesis. Secretory autophagy is a 'non-canonical' phenomenon in which autophagosome cargo is released from the cell through the plasma membrane by a lysosome-independent pathway[37]. Synaptotagmin Like 1 (SYTL1) protein, which is a critical regulator of secretion and exocytosis in cells[38], was another binding partner of AKT2 identified through our high-throughput human protein–protein interaction study (Supplementary Table 2). Therefore, we examined whether secretory autophagy is activated by increased AKT2 signaling. Using a pulldown assay, we confirmed that AKT2 associates with SYTL1 (Supplementary Fig. 9a) and forms a complex also containing tripartite motif-containing protein 16 (TRIM16)[37] and synaptosome-associated protein 23 (SNAP23)[37]. We further observed that this complex was needed for the efficient release of cellular cargo from cells during secretory autophagy (Supplementary Fig. 9b). Intriguingly, the formation of this secretory autophagy complex was inhibited by overexpressing SIRT5 (Supplementary Fig. 9b), potentially due to the regulation of AKT2 by SIRT5.

AKT2 overexpression in ARPE19 cells led to upregulation of other secretory autophagy mediators including FK-506-binding protein 51 (FKBP51)[39] and SNAP23, which was not seen in cells overexpressing AKT2+SIRT5 or a mutant form of AKT2 (K14A/R25E)[40] that has reduced kinase activity (Fig. 4a). Moreover, we observed that GFP-LC3-positive autophagosomes do not fuse with lysosomes in AKT2 overexpressing ARPE19 cells, suggesting that autophagosome clearance is impaired (Fig. 4b). This observation was confirmed using total internal reflection fluorescence (TIRF)[41] microscopy studies which demonstrated that the autophagosomes fused with the plasma membrane, but not with lysosomes (Supplementary Movies 1 and 2 and Supplementary Fig. 9c).

The conditioned medium from *Akt2* KI RPE explants isolated from aged (10-month-old) mice had significantly increased major secretory autophagy cargo[37], such as interleukin-1 beta (IL-1β) and high mobility group box 1 (HMGB1), compared to RPE cells isolated from age-matched wild-type mice (Supplementary Fig. 9d). A similar increase was observed in the levels of these cargo proteins in the conditioned medium from *CFH* (H/H) compared to *CFH* (Y/Y) iPSC-RPE cells (Supplementary Fig. 9e). Secretory autophagy and canonical degradative autophagy are integrated and highly regulated processes that share similar molecular mechanisms[37]. Accordingly, we observed that canonical autophagy was also reduced in *Akt2* KI RPE cells (Fig. 4c–e). RNAseq analysis of *Akt2* KI RPE cells identified a cluster of down-regulated genes involved in autophagosome formation[42], including autophagy-related 9B (*Atg9b*) and the Unc-51-like kinase 1 (*Ulk1*) (Fig. 4c). Their reduced protein abundance as well as accumulation of the autophagosome marker, p62[21], was confirmed by western blots (Fig. 4d). Genes involved in autophagosome formation are transcriptionally regulated by TFEB. Using chromatin immunoprecipitation (ChIP)[13], we found that TFEB fails to bind to the promoter of *Atg9b* in *Akt2* KI RPE cells (Fig. 4e). These abnormalities correlated with a decline in autophagy flux[21,43] in RPE cells overexpressing AKT2 (Fig. 4f, g). Moreover, double membranous autophagosomes[43]

accumulated in the RPE of *Akt2* KI mice by 15 months of age but were not seen in age-matched WT mice (Fig. 4h). Collectively, these results suggest that AKT2 upregulation in the RPE induces alterations in secretory autophagy that promote the release of material that could contribute to drusen formation.

## Targeting TFEB could assuage the lysosome-autophagy axis in RPE cells from mouse models and iPSC-derived RPE cells from CFH Y402H donors

Based on our findings, we speculated that targeting the AKT2/TFEB pathway could prevent early changes observed in AMD pathogenesis. We therefore targeted AKT2-dependent TFEB signaling in *CFH* (H/H) iPSC-derived RPE cells with (1) AAV2-TFEB-S467A, a mutation in the AKT-target residue of TFEB that induces constitutive nuclear localization[8], (2) treatment with the disaccharide trehalose, which activates TFEB independent of mTOR signaling[44], or (3) correction with the protective *CFH* I62V mutation[18]. These treatments rescued the abundance and activities of lysosomal and autophagy mediators Cathepsin D and L (Fig. 5a–e). To further prove the influence of complement on AKT2-dependent regulation of lysosomal function, we cultured iPSC-derived *CFH* (H/H) and (Y/Y) RPE cells with complement-competent human serum (CCHS), which provides complement factors including anaphylatoxins to mimic the age-induced increase in complement activation observed in AMD eyes[29], and then treated cells with an AKT2 inhibitor[22] and trehalose. Our results showed that CCHS treatment could deregulate lysosomal proteins in *CFH* (Y/Y) and (H/H) cells, and that both AKT2 inhibition and trehalose could rescue the levels of major lysosomal mediators CTSD, and lysosomal membrane protein LAMP1[5] in CFH (H/H) cells (Fig. 5f). Furthermore, lipid and APOE accumulation are involved in AMD pathogenesis, which can be observed in an in vitro AMD-like model system[29]. To ascertain the efficacy of AKT2 signaling in rescuing these features in vitro, we treated *CFH* (H/H) RPE cells grown in CCHS with the AKT2 inhibitor and found significantly diminished lipid accumulation by BODIPY staining, and a trend in the decrease of APOE accumulation (Supplementary Fig. 10a, b). These results indicate that AKT2-dependent regulation of lysosomal function could be a possible target for nullifying the disease-related phenotype observed in the "disease in a dish model"[29]. To examine if patient-derived Y402H RPE cells with lysosomal dysfunction could be rescued by targeting AKT2 and trehalose, we treated iPSC-derived RPE cells from AMD patients from *CFH* (Y/Y) and (H/H) donors (Supplementary Table 3) with or without trehalose or the AKT2 inhibitor. Our results showed that both trehalose or the AKT2 inhibitor could rescue Cathepsin D and LAMP1 in the *CFH* (H/H) cells, relative to untreated *CFH* (H/H) cells (Fig. 5g, h). Interestingly, the differences in these lysosomal mediators were not observed between the *CFH* (Y/Y) and (H/H) cells derived from human AMD donors (Fig. 5g, h), probably due to a coincident age-dependent decline in lysosomal function[5]. These results indicate that targeting TFEB could rejuvenate lysosomal function in *CFH* Y402H cells, which could be helpful in rescuing the disease phenotype in patients harboring this genetic variant. Further, trehalose rescued CLEAR gene expression and autophagy flux in RPE explants from *Cryba1* KO mice (Supplementary Fig. 11a–c), which show hyperactive mTORC1 and AKT2 signaling and a diminished TFEB-

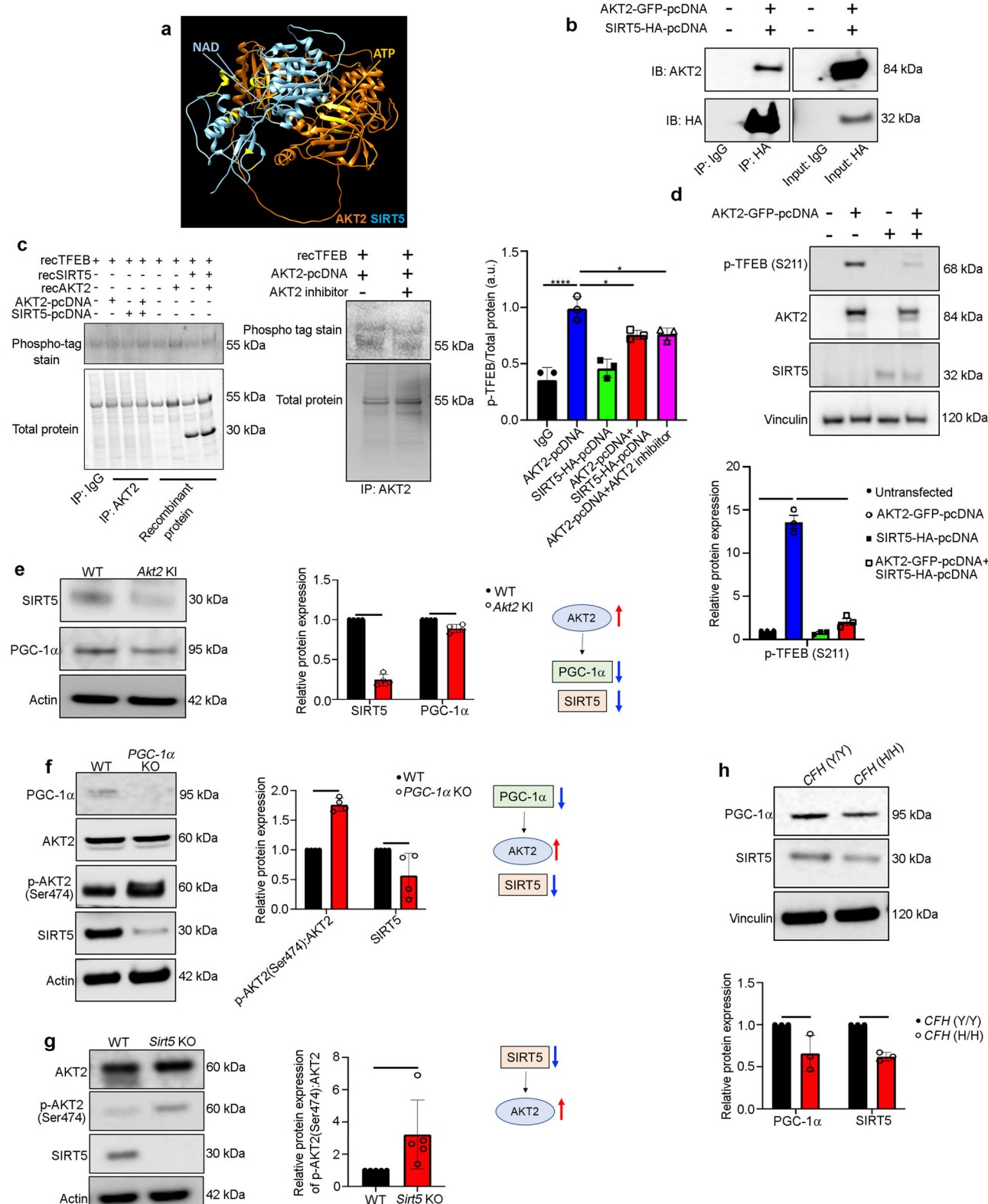

dependent autophagy/lysosome signaling axis in the RPE[13,21,45]. However, trehalose treatment did not influence the abnormal mTORC1 activation in the *Cryba1* KO RPE explants (Supplementary Fig. 11d). These findings suggest that trehalose can rejuvenate lysosomal function independent of mTOR signaling and encouraged us to investigate whether trehalose could prevent the lysosomal abnormalities observed in our mouse models[21,23] that develop an AMD-like phenotype. Trehalose administered[46] to 6-month-old *Cryba1* cKO or *Akt2* KI

mice for 3 consecutive months rescued CLEAR network gene expression including autophagy mediators, and restored lysosomal function in the RPE as well as the degenerative changes in the retina (Fig. 6a–g). We next explored whether targeting TFEB, independent of mTOR, using gene therapy could rescue TFEB nuclear translocation in *Akt2* KI RPE. Therefore, we cultured *Akt2* KI RPE explants and infected them with either AAV2-WT TFEB or AAV2-TFEB-S467A constructs, which showed that the S467A construct could induce nuclear localization of

**Fig. 3 | AKT2/SIRT5 signaling axis regulates TFEB activity. a** A hypothetical heterodimer of human AKT2/SIRT5 suggests a potential protein–protein interaction between these proteins. The ribbon structures of AKT2 and SIRT5 are colored orange and cyan, respectively. Residues related to the AKT2 and SIRT5 active sites are shown in yellow. The model suggests that both proteins have active sites with potential exposure to substrates and ligands, indicating that the AKT2/SIRT5 complex is catalytically active. **b** Co-immunoprecipitation study showing that SIRT5 and AKT2 are binding partners in SIRT5-HA-pcDNA and AKT2-GFP-pcDNA overexpressing ARPE19 cells, compared to untransfected controls upon pulldown with anti-IgG or anti-HA antibodies, respectively. n = 3. **c** Pulldown of AKT2 with anti-AKT2 antibody and subsequent incubation of the pulldown complex with TFEB (recombinant; rec) in vitro to evaluate phosphorylation of TFEB using the phospho-tag gel staining, showed increased p-TFEB/total TFEB in AKT2 overexpressing ARPE19 cells, compared to control. This TFEB phosphorylation was reduced when cells were overexpressing both AKT2 and SIRT5 or the AKT2 pull-down complex was incubated with an AKT2 inhibitor, indicating SIRT5 can modulate AKT2 activity and that AKT2 activity is critical in regulating TFEB phosphorylation. n (biological replicate) = 3. **d** Western blot showing that simultaneous transfection of AKT2 and SIRT5 pcDNAs could rescue the levels of p-TFEB (S211) in ARPE19 cells, relative to cells overexpressing only AKT2. AKT2 and SIRT5 overexpression in the ARPE19 cells was also confirmed by western blot. n (biological replicate) = 3. Western blot analysis showing reduced expression of SIRT5 and PGC-1α in RPE cells from **e** *Akt2* KI, **f** *PGC-1α* KO, and **g** *Sirt5* KO mice, as well as **h** iPSC-derived RPE cells from *CFH* (H/H) donors, compared to controls. n = 4 (mice) and n = 3 (iPSC cells). All values are Mean ± S.D. ****$P < 0.0001$, ***$P < 0.001$, **$P < 0.01$, *$P < 0.05$. The statistical tests used in (**c** and **d**) is One-way ANOVA followed by Tukey's post-hoc test whereas in (**e**–**h**) is Student's t-test (Two-sided). The exact p-values are **c** $P = 0.00002$ (AKT2-pcDNA vs IgG), $P = 0.0001$ (AKT2-pcDNA+SIRT5-HA-pcDNA vs AKT2-pcDNA), $P = 0.0462$ (AKT2-pcDNA+ AKT2 inhibitor vs AKT2-pcDNA); **d** $P = 0.00062$ (AKT2-GFP-pcDNA vs untransfected), $P = 0.00051$ (AKT2-GFP-pcDNA+SIRT5-HA-pcDNA vs AKT2-pcDNA); **e** SIRT5: $P = 0.0021$ and PGC-1α: $P = 0.008$ (*Akt2* KI vs WT); **f** p-AKT2 (S474): AKT2: $P = 0.000832$ and SIRT5: $P = 0.03$ (*PGC-1α* KO vs WT); **g** $P = 0.0079$ (*Sirt5* KO vs WT); **h** PGC-1α: $P = 0.0252$ and SIRT5: $P = 0.0057$ (*CFH* (H/H) vs *CFH* (Y/Y)). Source Data is provided in the Source Data file.

TFEB compared to WT TFEB construct (Supplementary Fig. 12). Moreover, while retinal degenerative changes are typically seen in both these mouse models by 9–10 months of age[21,22], they were not visible in trehalose-treated mice (Fig. 6h, i). Thus, augmenting TFEB rescued autophagy and lysosomal biogenesis in both *CFH* (H/H) iPSC-derived RPE cells and mouse models with lysosomal deficits (Fig. 6j). Importantly, the rescue of lysosomal dysfunction prevented both RPE degeneration and basal laminar deposit formation in our mouse AMD model, indicating that targeting TFEB could be of importance in delaying the progression of this blinding eye disease.

## Discussion

The RPE is a single layer of epithelium located above Bruch's membrane and the choriocapillaris and below the retinal photoreceptors. The RPE serves as a conduit for oxygen, nutrients, and waste products between the outer retina and the choriocapillaris[6]. It is accepted that the RPE plays an important role in AMD pathogenesis. Nonetheless, contributions from the photoreceptors, Bruch's membrane, and the choriocapillaris should not be neglected[1,6]. Indeed, it has been reported that early/dry AMD is characterized by attenuation of the choriocapillaris, which provides oxygen and nutrients for both the RPE and photoreceptors and is essential to their survival[1,6]. Several dysfunctional pathways including impaired lysosomal function in the RPE have been shown to contribute to AMD pathobiology[1,2,5,6,37]. Herein, we find that *Akt2* KI mice that overexpress AKT2 specifically in the RPE but not the retina, develop an AMD-like phenotype including RPE degeneration, basal deposit formation, and impaired retinal function, which are mitigated by restoring lysosomal function with trehalose. Importantly, we mechanistically connect increased AKT2 signaling in the RPE with lysosomal dysfunction through an impaired AKT2-SIRT5-TFEB/E3-PGC-1α signaling pathway, which contributed to RPE degeneration. The induction of secretory autophagy owing to abnormal lysosomal function likely contributes to basal deposit formation. To provide relevance to AMD in humans, we show that AKT2 signaling is increased to impair lysosomal function in the RPE of AMD donor eyes and iPSC-derived RPE cells that harbor the high-risk *CFH* 402H variant.

TFEB and TFE3, members of the MITF transcription factor family, tightly control lysosomal biogenesis and function as well as autophagy at the level of transcription[7,45]. A decline in TFEB can be compensated by a concomitant increase in TFE3[25]. These transcription factors are regulated by mTOR-dependent and independent signaling pathways[7–10]. AKT1 can regulate TFEB through mTOR-independent signaling[8]. However, the regulatory role of AKT2 on TFEB/E3 and their compensatory regulation in the context of AMD pathogenesis has been unclear. AKT2 is a known upstream regulator of mTORC1 and a downstream mediator for PDK1 and mTORC2[11,12,23]. However, the mechanism of AKT2 activation remains elusive and involves several pathways and multiple phosphorylation sites (e.g., T309 and S474) that collectively govern AKT2 activity in cells depending on specific conditions[11,12,23,40]. Nonetheless, as it has been reported that S474 is associated with maximal AKT2 activity in several cell types[12,40], we examined the levels of AKT2 S474 phosphorylation to understand its role in regulating lysosomal function in RPE cells. Additionally, we demonstrated that AKT2 upregulation in *Tfeb* null MEF cells prevented a compensatory increase in TFE3 that subsequently decreased CLEAR network gene expression to impair lysosomal function. We also discovered that AKT2 can bind SIRT5, which can mitigate lysosomal dysfunction induced by increased AKT2 signaling. Intriguingly, we find that increased AKT2 signaling inhibits PGC-1α, a major transcription factor for mitochondrial biogenesis[32–34] that downregulates mitochondrial biogenesis genes including SIRT5[33]. Besides mitochondrial regulation, SIRT5 can regulate both lysosomal function and autophagy[30,31]. Our results suggest that SIRT5 and AKT2 co-regulate each other through PGC-1α, and thereby modulate TFEB-mediated lysosomal and mitochondrial function in RPE cells, providing an important functional connection between the mitochondria and lysosomes in maintaining RPE homeostasis. We postulate that increased AKT2 in the RPE leads to downregulation of both PGC-1α and SIRT5 to limit TFEB activity and subsequent lysosomal and mitochondrial function (Fig. 6g). This connection has pathogenic relevance given that both mitochondrial and lysosomal dysfunction are recognized as etiologic factors in early AMD.

AMD is a multi-factorial disease caused by a combination of genetic and environmental risk factors[1,15]. The identification of complement factor alleles that elevate AMD risk has been well documented[18,19]. The *CFH-CFHR5* region that includes the *CFH* Y402H variant on chromosome 1q32 (Chr1 locus) is a major loci associated with significant disease risk[18,19] while the *CFH* I62V variant confers protection[18]. However, AMD is not a monogenic disease and the higher risk associated with the *CFH* polymorphism does not reflect an immediate cause for AMD. Rather, this polymorphism, which leads to less effective inhibitory control of the alternative pathway, is likely to result in the accumulation of active complement components (e.g., C3a or C5a) which, in turn, activate downstream signaling molecules that contribute to disease onset[18,19,29]. It was previously reported that *CFH* Y402H-AMD-patient-specific RPE cells have a significant increase in the number of swollen lysosome-like vesicles with fragile membranes, Cathepsin D leakage into drusen-like deposits, reduced lysosomal function, diminished autophagy, and elevated mTORC1 signaling[20,29]. However, the underlying mechanism of lysosomal dysfunction in *CFH* Y402H harboring RPE cells has been unknown. We now provide a mechanism to explain these observations. In AMD donor globes, AKT2 is upregulated in macular RPE[13,14] and can induce abnormalities in the autophagy-lysosome cellular axis by

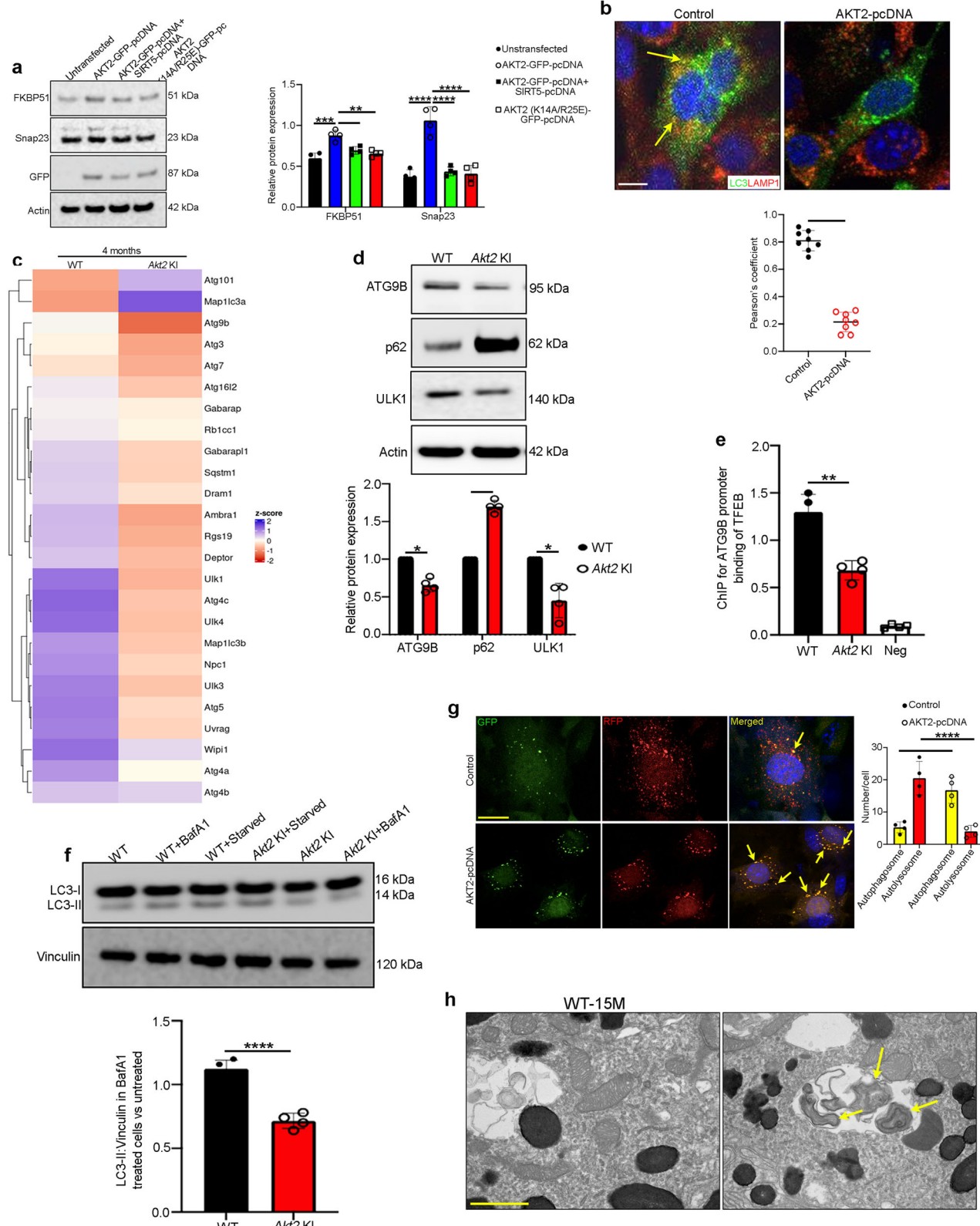

deregulating TFEB signaling, independent of mTORC1[9]. Likewise, we find that AKT2 signaling is increased in iPSC-derived RPE cells with the *CFH* (Y402H) risk allele[29], which decreases lysosomal function. The decrease in the major lysosomal hydrolases[5] Cathepsin D and L in the RPE from AMD donors and in individuals harboring the *CFH* risk allele indicates that the protective function of *CFH* on lysosomal function in these RPE cells is compromised.

While autophagy is required to prevent the accumulation of non-functional misfolded proteins, it is also known to cause disease when overactive[37]. We further discovered that AKT2 is a binding partner for SYTL1, a critical regulator of secretion and exocytosis in cells[38], a phenomenon thought to be critical in AMD pathogenesis[37,39]. Specifically, secretory autophagy was reported to contribute to the secretion of inflammatory mediators and drusen components by the

**Fig. 4 | Akt2 upregulation inhibits macroautophagy and triggers secretory autophagy in RPE cells. a** Western blot showing elevated levels of FKBP51 and Snap23 in AKT2 overexpressing ARPE19 cells, which were reduced upon simultaneous upregulation of SIRT5 or overexpression of an AKT2 inactive mutant (K14A/R25E). n (biological replicate) = 4. **b** Immunofluorescence assay showing association of LC3-positive autophagosomes (green) with lysosomes (Lamp1-positive; red) in control ARPE19 cells (arrows in **b**), which was significantly reduced upon AKT2 overexpression. Scale bar = 50 μm. n (biological replicate) = 8. **c** RNAseq analysis from RPE cells of 4-month-old WT and *Akt2* KI mice showing differential expression of several autophagy-related genes. n = 3. **d** Western blot showing reduced expression of autophagy mediators ATG9B and ULK1, as well as upregulation of the autophagosome marker p62/SQSTM1 in *Akt2* KI RPE cells, relative to WT. n (biological replicate) = 4. **e** Chromatin immunoprecipitation showing diminished binding of TFEB on Atg9b promoter region in *Akt2* KI RPE cells, compared to WT. n (biological replicate) = 4. **f** Western blot showing reduced autophagy flux (Ratio of LC3-II/ Vinculin in BafA1 treated vs untreated) in *Akt2* KI RPE explants compared to WT when treated with Bafilomycin A1 (BafA1; 1 μm) for 4 h. n (biological replicate) = 4. **g** ARPE19 cells were transfected with AKT2 construct for 48 h or left untreated (control), followed by an overnight infection with an Adenovirus-GFP-RFP-LC3B construct to label the autophagosomes (yellow) and autolysosomes (red). The number of autolysosomes (red puncta) was significantly decreased in AKT2 overexpressing cells (arrows in **c**) when compared to controls, suggesting a decline in autophagy flux. Scale bar = 50 μm. n (biological replicate) = 4. **h** Transmission electron micrographs showing double membranous autophagosomes in the RPE cells of 15-month-old *Akt2* KI mice, but not in age-matched WT. Scale bar = 600 nm. n = 5. All values are Mean ± S.D. ****$P < 0.0001$, ***$P < 0.001$, **$P < 0.01$, *$P < 0.05$. The statistical tests used in (**a, c,** and **f**) is One-way ANOVA followed by Tukey's post-hoc test whereas in (**b, d, g**) is Student's t-test (Two-sided). The exact p-values are **a** FKBP51: $P = 0.0007$ (AKT2-GFP-pcDNA vs untransfected), $P = 0.0327$ (AKT2-GFP-pcDNA+SIRT-HA-pcDNA vs AKT2-GFP-pcDNA), $P = 0.0081$ (AKT2(K14A/R25E)-GFP-pcDNA vs AKT2-GFP-pcDNA); **b** $P = 1.598E-10$ (AKT2-pcDNA vs Control); **d** ATG9B: $P = 0.0139$, p62: $P = 0.0013$, ULK1: $P = 0.017$; **e** $P = 0.001$ (*Akt2* KI vs WT); **f** $P = 9.45202E-05$ (*Akt2* KI vs WT); **g** Autophagosome: $P = 0.0012$, Autolysosome: $P = 0.000082$ (AKT2-pcDNA vs Control). Source Data is provided in the Source Data file.

RPE[37]. We find that overexpression of SIRT5 inhibited secretory autophagy mediators that were formed due to AKT2 and SYTL1 binding. In addition to secretory autophagy, RPE cells can release extracellular vesicles (EVs) that contain proteins identified in drusen including those involved in oxidative stress, complement, and inflammation, likely due to altered lysosomal/autophagic function[47]. Furthermore, these RPE cells release these EVs containing drusen-associated proteins basally in response to AMD stressors[47]. These and our findings highlight the role of RPE-derived secretory products in drusen formation.

iPSC-derived RPE cells homozygous for the *CFH* Y402H risk allele have been reported to develop an AMD-like phenotype in an in vitro model system upon treatment with serum-derived anaphylatoxins (containing activated complement), while heterozygotes did not[29]. Moreover, it was shown that CFH-H/H (*CFH* Y402H variant on a *cfh*⁻/⁻ background) mice have elevated serum levels of CFH, which is similar to human AMD patients, develop an AMD-like phenotype consisting of vision loss, loss of RPE function, and increased basal laminar deposits following feeding with a cholesterol-enriched diet[48]. Conversely, while *cfh*⁻/⁻ (*CFH* null) mice also manifest decreased visual acuity compared to age-matched controls and develop C3 accumulation in the RPE, they did not show autophagy/lysosomal dysfunction or visible photoreceptor or RPE degeneration despite aging mice for up to 24 months[49]. Using our model of iPSC-derived RPE cells from human donors, we report here that deleting *CFH* also did not affect lysosomal function. Collectively, these results suggest that patients with the high-risk *CFH* variant may develop AMD due to altered extracellular and intracellular CFH protein which accumulates with age; consequently, the complete loss of the CFH protein fails to develop any affect[50]. We speculate that accumulation of altered CFH protein leads to activation of signaling pathways, which results in abnormal lysosomal-mediated clearance, lipid accumulation as well as chronic inflammation, to promote the development of AMD.

The most popular therapeutic target to regulate TFEB/TFE3 nuclear localization and lysosomal activity has been mTORC1. However, these studies have been unsuccessful because of unacceptable side effects[51]. Given the lysosomal defects in dry AMD[5,21,37,45], particularly in patients who harbor the Y402H *CFH* risk variant[20], and important interactions between lysosomes and mitochondria, our results provide an alternative therapeutic strategy in which we can target multiple, specific relevant downstream signaling pathways. We propose that activating the lysosome/autophagy pathway, and its communication with mitochondria, represents a potential therapeutic target that could rejuvenate two key organelles that become impaired in the RPE of patients with dry AMD.

## Methods

### Antibodies
The primary antibodies AKT2 (3063S), p-AKT2 (8599S), SIRT5 (8782S), CTSD (69854S), ULK1 (8054T), FKBP51 (8245S), p-TFEB (37681S), and GFP (2555S) were purchased from Cell Signaling Technology. CTSL (NB100-1775), p62/SQSTM1 (NBP1-42821), and PGC-1α (NBP1-04676) were purchased from Novus Biologicals. TFE3 (14480-1-AP), ATG9B (PA5-20998), Ezrin (PA518541), Trim16 (PA5-110515), and Snap23 (PA1-738) were purchased from Thermo Fisher. Loading controls were vinculin (ab129002, Abcam) and beta-Actin (4970S, Cell Signaling). TFEB (A303-673A) and SYTL1 (A305-648A-T) antibodies were purchased from Bethyl Laboratories. The p-TFE3 antibody was a gift from Dr. Rosa Puertollano's laboratory. The secondary antibodies used in this study were purchased from KPL: anti-rabbit (074–1506) and anti-mouse (074–1806).

### Animals and trehalose treatment
All animal studies were conducted in accordance with the Guide for the Care and Use of Animals (National Academy Press) and were approved by the University of Pittsburgh Animal Care and Use Committee. The authors have complied with all relevant ethical regulations for animal maintenance and euthanasia during this study. Both male and female RPE-specific *Akt2* KI, *Akt2* cKO, and βA3/A1-crystallin conditional (*Cryba1* cKO), as well as βA3/A1-crystallin complete knockout (*Cryba1* KO), PGC-1α KO, and *Sirt5* KO mice were generated as previously described[21–23,34,35]. All mice that were used in this study were negative for RD8 mutation[21]. At 6 months of age, the *Cryba1* cKO and *Akt2* KI mice were injected with trehalose (T9449-100G, Sigma Aldrich) intraperitoneally at 1 mg/g/day for 2 days and then given 2% trehalose (w/v) in drinking water *ad libitum*[46] for the entire experimental duration of 3 months.

### iPSC-derived cells culture
Human iPSC-derived RPE cells were obtained from genetically screened human donors as explained previously[29,52,53]. The donors were randomly chosen and from both sexes to nullify any gender-based bias. Both the control, *CFH* (Y/Y), and the risk allele *CFH* (H/H) containing donor cells were differentiated and maintained as described previously[29,52,53]. *CFH* (H/H) cells with the rescue *CFH* I62V mutation and *CFH* null cells were also generated and maintained as described earlier. The iPSC lines were cultured on vitronectin-coated plates in E8 complete medium (A1517001, Thermo Fisher)[29,52,53]. The recombinant vitronectin (A14701SA, Thermo Fisher) was diluted in DPBS at the concentration of 1:200 to get the final concentration of 2.5 mg/ml. The cells were passaged every 4–5 days at 70–80% confluency[29,52,53] and were fed every 24 h. The differentiated cells were

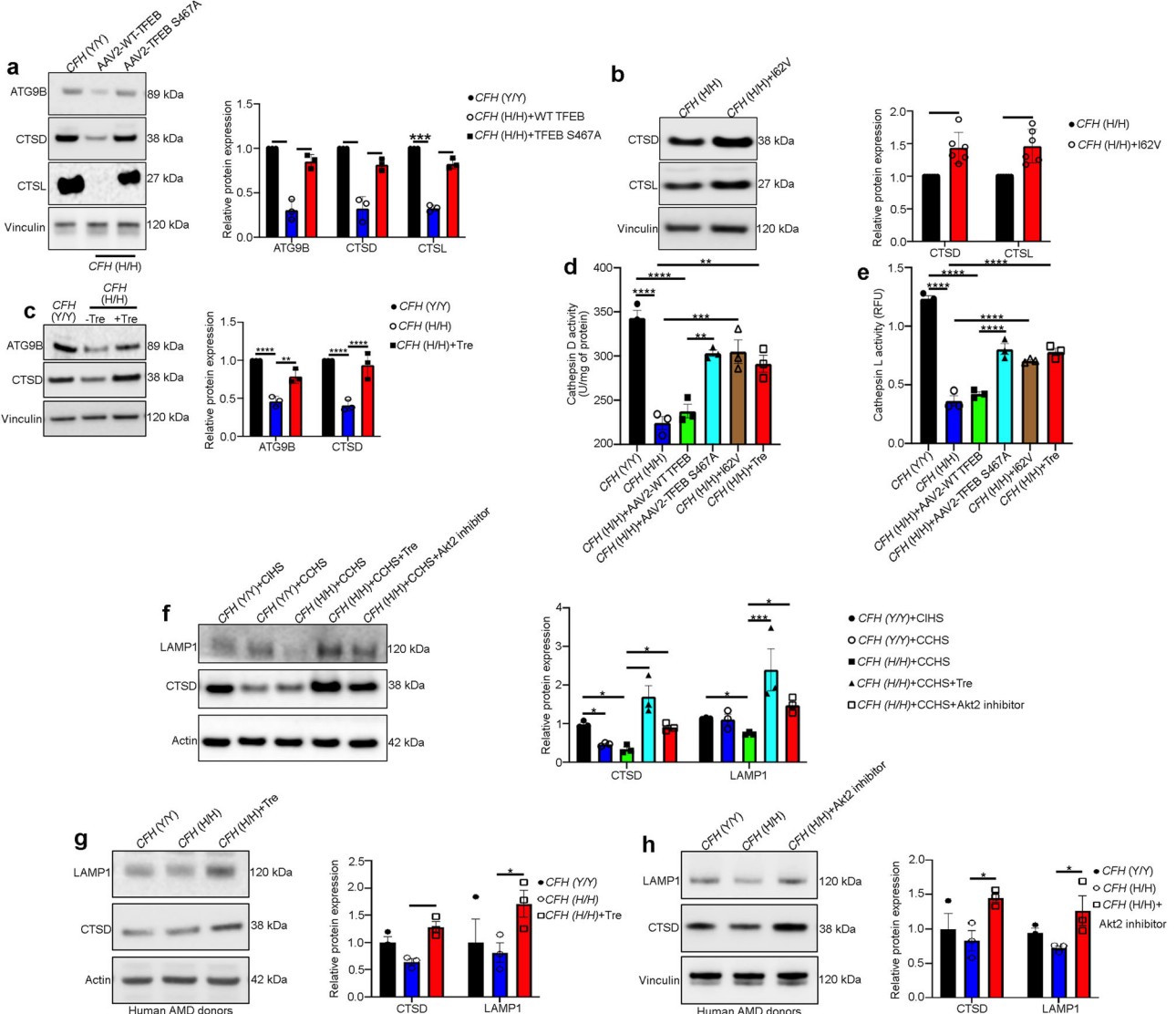

**Fig. 5 | Targeting TFEB independent of mTOR can rejuvenate lysosomal function in CFH Y402H iPSC-derived RPE cells. a** Western blot analysis showing that in iPSC-derived RPE cells from *CFH* (H/H) risk allele-containing donors, over-expression of mutant TFEB-S467A construct ($10^6$ vg/ml for 48 h) or **b** correction with *CFH* I62V mutation or **c** treatment with trehalose (100 mM for 20 h) rescued the levels of the lysosomal hydrolases Cathepsin D, Cathepsin L and the autophagosome mediator ATG9B, compared to untreated *CFH* (H/H) cells. n (biological replicate) = 3 (TFEB-S467A and trehalose groups), n (biological replicate) = 6 (*CFH* I62V rescue). **d, e** Cathepsin D and cathepsin L activities also showed significant rescue upon TFEB-S467A infection, I62V correction, or trehalose treatment to *CFH* (H/H) cells. n (biological replicate) = 3. **f** Trehalose (100 mM for 48 h) or AKT2 inhibitor (10 nM for 48 h) treatment to iPSC-derived RPE cells from CFH (H/H) donors incubated with CCHS (5% for 48 h) or **g, h** iPSC-derived RPE cells from human AMD donors with CFH (H/H) genotype, rescued the levels of Lamp1 and cathepsin D, compared to untreated CFH (H/H)+CCHS or CFH (H/H; from AMD donors) cells. n (biological replicate) = 3. All values are Mean ± S.D. *P < 0.05, ****P < 0.0001, ***P < 0.008, **P < 0.01, *P < 0.05. The statistical tests used in (**a, c, d–h**) is One-way ANOVA followed by Tukey's post-hoc test, and in (**b**) is Student's t-test (Two-sided). The exact p-values are **a** ATG9B: P = 0.0167, CTSD:

P = 0.0234, CTSL: P = 0.0006 (CFH (H/H)+WT TFEB vs CFH (Y/Y)), ATG9B: P = 0.0085, CTSD: P = 0.0235, CTSL: 0.0012 (CFH (H/H)+TFEB-S467A vs CFH (H/H) +WT TFEB); **b** CTSD: P = 0.0067, CTSL: P = 0.007 (CFH (H/H)+I62V vs CFH (H/H)); **c** ATG9B: P = 0.00002, CTSD: P = 0.00001 (*CFH* (H/H) vs *CFH* (Y/Y)) and ATG9B: P = 0.0020, CTSD: 0.00003 (*CFH* (H/H)+Tre vs *CFH* (H/H)). **d** P = 0.00001 (*CFH* (H/ H) vs *CFH* (Y/Y)), P = 0.00003 (*CFH* (H/H)+AAV2-WT TFEB vs *CFH* (Y/Y)), P = 0.00225 (*CFH* (H/H)+AAV2-TFEB-S467A vs *CFH* (H/H)+AAV2-WT TFEB), P = 0.00037 (*CFH* (H/H)+I62V vs *CFH* (H/H)), P = 0.00191 (*CFH* (H/H)+Tre vs *CFH* (H/ H)); **e** P = 0.00001 (*CFH* (H/H) vs *CFH* (Y/Y)), P = 0.000001 (*CFH* (H/H)+AAV2-WT TFEB vs *CFH* (Y/Y)), P = 0.00002 (*CFH* (H/H)+AAV2-TFEB-S467A vs *CFH* (H/H) +AAV2-WT TFEB), P = 0.00008 (*CFH* (H/H)+I62V vs *CFH* (H/H)), P = 0.00001 (*CFH* (H/H)+Tre vs *CFH* (H/H)). **f** CTSD: P = 0.0401 (*CFH* (Y/Y)+CCHS vs *CFH* (Y/Y)+CIHS), P = 0.0478 (*CFH* (H/H)+CCHS vs *CFH* (Y/Y)+CIHS), P = 0.0011 (*CFH* (H/H)+CCHS +Tre vs *CFH* (H/H)+CCHS), LAMP1: P = 0.049 (*CFH* (Y/Y)+CCHS vs *CFH* (Y/Y)+CIHS), P = 0.0003 (*CFH* (H/H)+CCHS+Tre vs *CFH* (H/H)+CCHS). **g** CTSD: P = 0.0031, Lamp1: P = 0.0452 (*CFH* (H/H)+Tre vs *CFH* (H/H)); **h** CTSD: P = 0.0263, Lamp1: P = 0.0491 (CFH (H/H)+Akt2 inhibitor vs CFH (H/H)). Source Data is provided in the Source Data file.

confirmed for RPE markers and were subjected to downstream experiments. To ascertain the contribution of systemic complement factors on the lysosomal function in CFH (Y/Y) and (H/H) cells, these cells were treated with CIHS (5%) and CCHS (5%)[29] with or without treatment with either AKT2 inhibitor at a dose of 10 nM (CCT12890;

S2635, Selleckchem) or Trehalose (T0167, Sigma Aldrich) at a dose of 100 mM for 48 h before collecting samples for western blot and immunofluorescence studies.

The derivation of iPSC cells from conjunctival cells from human AMD donors and the differentiation of iPSC into RPE was performed as

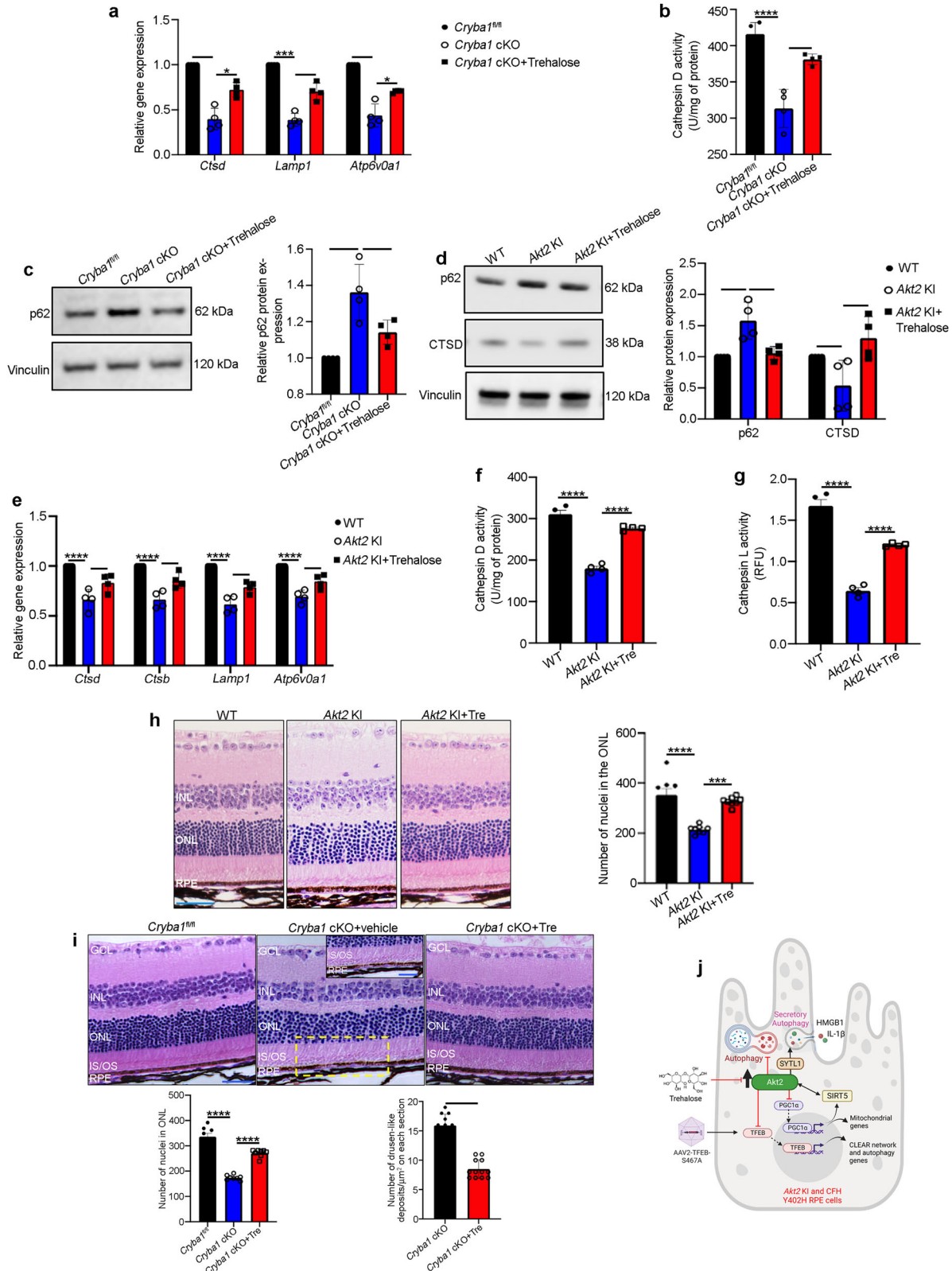

previously described[54]. Briefly, the iPSC cells were at passage 4 during the experiments. In the first week, the medium used was 5% FBS +MEMα + Rock inhibitor (working concentration 10 µM). During the second week, the medium that was used was 2% FBS+MEMα + Nicotinamide (working concentration 1 µM). The media was changed thrice per week until cells became pigmented and showed cobblestone morphology as explained previously[54]. Differentiated RPE cells were

cultured for at least one week before being treated with either AKT2 inhibitor at a dose of 10 nM (CCT12890; S2635, Selleckchem) or Trehalose (T0167, Sigma Aldrich) at a dose of 100 mM for 48 h.

### Human AMD donors
The human AMD donor sections were purchased from Lion's Gift of Sight, Minnesota. The control (n = 5) and AMD (n = 3) donors were all

**Fig. 6 | Trehalose rejuvenates lysosomal function and rescues autophagy in mouse models in vivo.** RPE cells from 6-month-old *Cryba1* cKO mice treated with trehalose for 3 consecutive months show significant rescue in the levels of **a** CLEAR network genes like *Ctsd*, *Lamp1* and *Atp6v0a1*, **b** cathepsin D (CTSD) activity and **c** protein levels of the autophagosome marker p62/SQSTM1, relative to RPE cells from vehicle (water) treated *Cryba1* cKO mice. n (biological replicate) = 4. RPE cells from 6-month-old *Akt2* KI mice treated with trehalose for 3 consecutive months showed statistically significant rescue of **d** protein levels of autophagosome marker p62/SQSTM1 and lysosomal hydrolase, CTSD as well as **e** CLEAR network gene (*Ctsd*, *Ctsb*, *Lamp1*, *Atp6v0a1*) expression, compared to vehicle-treated *Akt2* KI mice. n (biological replicate) = 4. Trehalose treatment to *Akt2* KI mice showed noticeable rescue in **f** CTSD and **g** CTSL activities relative to untreated *Akt2* KI mice. n (biological replicate) = 4. Hematoxylin-eosin stained sections and quantification showing a decrease in ONL thickness in **h** *Akt2* KI and **i** *Cryba1* cKO mice, which was rescued upon trehalose treatment. Prevalence of a patchy RPE layer with the accumulation of drusen-like deposits (asterisks in **i** and inset) in *Cryba1* cKO retina (vehicle control), with rescue by trehalose treatment. Scale bar = 100 µm (inset = 20 µm). n (biological replicate) = 5. **j** Created with BioRender.com released under a Creative Commons Attribution-NonCommercial-NoDerivs 4.0 International license https://creativecommons.org/licenses/by-nc-nd/4.0/deed.en. Cartoon showing that AKT2 upregulation in RPE cells from *Akt2* KI mice and *CFH* (H/H) iPSC-derived

cells trigger deregulation of SIRT5/PGC-1α dependent lysosomal function. Trehalose and S467A TFEB mutant construct rescued the lysosomal abnormalities and delayed the progression of early RPE changes in a mouse model. All values are Mean ± S.D. ****P < 0.0001, ***P < 0.001, **P < 0.01, *P < 0.05. The statistical test used in (**a**–**i**) is One-way ANOVA followed by Tukey's post-hoc test. The exact p-values are **a** *Ctsd*: P = 0.0042, *Lamp1*: P = 0.0010, *Atp6v0a1*: P = 0.0062 (*Cryba1* cKO vs *Cryba1*^fl/fl^), *Ctsd*: P = 0.0129, *Lamp1*: P = 0.0055, *Atp6v0a1*: P = 0.0489 (*Cryba1* cKO +Trehalose vs *Cryba1* cKO); **b** P = 0.00006 (*Cryba1* cKO vs *Cryba1*^fl/fl^), P = 0.0013 (*Cryba1* cKO+Trehalose vs *Cryba1* cKO); **c** P = 0.0013 (*Cryba1* cKO vs *Cryba1*^fl/fl^), P = 0.0258 (*Cryba1* cKO+Trehalose vs *Cryba1* cKO); **d** p62: P = 0.0037 and CTSD: P = 0.0461 (*Akt2* KI vs WT), p62: P = 0.0072 and CTSD: P = 0.0092 (*Akt2* KI+Trehalose vs *Akt2* KI); **e** *Ctsd*: P = 0.00001, *Ctsb*: P = 0.00002, *Lamp1*: P = 0.000035, *Atp6v0a1*: P = 0.000029 (*Akt2* KI vs WT), *Ctsd*: P = 0.0020, *Ctsb*: P = 0.0005, *Lamp1*: P = 0.0017, *Atp6v0a1*: P = 0.0056 (*Akt2* KI+Trehalose vs *Akt2* KI); **f** P = 0.00002 (*Akt2* KI vs WT), P = 0.000042 (*Akt2* KI+Trehalose vs *Akt2* KI); **g** P = 0.000049 (*Akt2* KI vs WT), P = 0.00006 (*Akt2* KI+Trehalose vs *Akt2* KI). **h** P = 0.00001 (*Akt2* KI vs WT), P = 0.0001 (*Akt2* KI+Trehalose vs *Akt2* KI); **i** ONL thickness: P = 0.000027 (*Cryba1* cKO vs *Cryba1*^fl/fl^), P = 0.000019 (*Cryba1* cKO +Trehalose vs *Cryba1* cKO), drusen-like deposits: P = 0.0005 (*Cryba1* cKO+Trehalose vs *Cryba1* cKO; Student's t-test). Source Data is provided in the Source Data file.

Caucasians with an average age of 76 ± 5 years. The control donors did not have any eye diseases. The AMD donors (both sexes were included) were staged according to the Minnesota Grading System (MGS) and all the donors were classified as MGS2, as explained earlier[13].

## RPE flatmount and immunostaining
After euthanizing WT and *Akt2* KI mice, the eyes were immediately enucleated using angled forceps. Eyes were briefly washed in 1× PBS at RT and dabbed on Kimwipes to remove PBS. The eyes were then introduced into tubes containing 4% paraformaldehyde (PFA). After 2 h of fixation eyes were cleaned by trimming away the extra-ocular tissue and debris under the dissecting microscope, leaving the optic nerve intact[22,24]. The cleaned eyes were again added to the tubes containing PFA and left for another 3 h at RT. The fixed eyes were removed from PFA, added to 30% sucrose, and for 10–12 h at 4 °C. The eyeballs were then dissected under the dissecting microscope for RPE flatmount preparation as described previously[22,24]. To eliminate auto-fluorescence from the melanin in the RPE flatmount, the RPE cups were treated with 1 mg/ml solution of sodium borohydride for 30 min. This was followed by permeabilization and blocking steps as previously described[22,24]. The flatmounts were incubated with antibodies to ZO-1 (1:200) (13663S, Cell Signaling Technology) or TFEB (A303-673A, Bethyl Laboratories; for immunofluorescence studies on RPE explant cultures) in primary antibody buffer (1% Tween 20 + 0.5% BSA in PBS) and incubated, overnight at 4 °C. The primary antibody was removed, and the RPE flatmounts were washed with PBS + 1% Tween 20. The flatmounts were then incubated with Anti-Rabbit Alexa fluor 555 secondary antibody (1:500) (A31572, Invitrogen) prepared in secondary antibody buffer (0.1% Tween 20 + 0.5% BSA in PBS), along with Hoechst (1:2000) (62249, Thermo Fisher), for nuclear staining and incubated in the dark for 2 h at RT[22,24]. The slides were then washed with 1× PBS, 5 times, for 15 min each and then mounted with DAKO mounting media before imaging using an Olympus IX81 confocal microscope[22,24].

## Immunofluorescence in iPSC-derived RPE cells
5–6-week-old fully matured, polarized iRPE monolayers were used for all the experiments. iRPE cells were treated for 48 h with 5% CCHS (S1-LITER, EMD Millipore) with media change every 24 h. 5% CIHS (heat-inactivated CCHS) acted as a control in these experiments. After the 48 h treatments, iRPE monolayers were fixed in 4% paraformaldehyde for 20 min at RT. After fixation, cells were blocked in Immuno-cytochemistry (ICC) buffer (1× PBS (10010-023, Thermo Fisher), 1% bovine serum albumin (BSA) (160069, MP Biomedicals), 0.25% Tween

20 (900-64-5, Affymetrix), 0.25% Triton X-100) (9002-64-5, Sigma) for 1 h at RT. APOE (1:1000) were diluted in ICC buffer, added to the cells, and incubated overnight at RT. Following the overnight primary antibody incubation, the cells or membranes were washed three times with ICC buffer. Secondary antibodies were diluted 1:1000 in ICC buffer, added to the cells or membranes, and incubated in the dark for one hour at RT. The cells and membranes were washed three times with ICC buffer and mounted on a glass slide with Fluoromount-G aqueous mounting medium (0100-01; Southern Biotech), and a glass coverslip. Samples were imaged using a Zeiss microscope[29].

For BODIPY staining, after the cells were fixed in PFA, BODIPY (D3922, Thermo Fisher.) was diluted at a 1:300 concentration in DPBS and cells were incubated overnight on a rocker at RT. Cells were washed 3 times with DPBS and mounted on a slide as mentioned above. The quantification of APOE and BODIPY was carried out as according to our recently published protocol[55].

## RPE explant culture
Immediately after euthanization of 3-month-old WT or *Cryba1* KO or *Akt2* KI mice, the eyes were enucleated, and anterior segments were removed. The posterior eye cups were then cut into four petals and the neural retina was carefully removed[24,43]. The harvested RPE-choroid-sclera (RCS) complex was placed onto PVDF membranes after flattening them by making several relaxing cuts. The explants were then cultured face-up in complete media as previously described[24,43]. Treatment with trehalose was done on *Akt2* KI or *Cryba1* KO RPE explants at a dose of 100 mM for 20 h as explained previously[24,28,43]. *Akt2* KI RPE explants were also infected with either AAV2-WT TFEB or AAV2-TFEB-S467A constructs at a dose of $10^5$ vg/ml for 24 h. The explants were then fixed and immunostained for TFEB, to ascertain nuclear translocation[22,24].

## Mouse retinal cryosectioning
Eyes from freshly euthanized WT and *Akt2* KI mice were fixed following the same procedure for RPE flatmount preparations[22,24]. The cryosectioning of the eye globes and subsequent permeabilization and blocking of the sections was performed as explained previously[22,24]. The slides were incubated with a primary antibody cocktail containing either rhodopsin (ab98887, Abcam) or EBP50 (PA1090, Thermo Fisher), in primary antibody buffer (1% Tween 20 + 0.5% BSA in PBS) at a dilution of 1:100 and incubated in a moist chamber overnight at 4 °C. The slides were washed with PBS + 1% Tween 20 three times. Secondary antibody cocktail Anti-mouse Alexa fluor 647 (1:200) and Anti-Rabbit

Alexa fluor 555 (1:200) was prepared in secondary antibody buffer (0.1% Tween 20 + 0.5% BSA in PBS) also containing Hoechst (1:2000) and Alexa fluor 488-phalloidin (A12379, Thermo Fisher) (1:1000) and was added to each slide covering the sections completely[22,24]. The slides were washed with 1× PBS, 5 times, for 15 min each and then mounted with a coverslip using DAKO mounting medium until imaged with an Olympus IX81 confocal microscope[22,24]. The length of phalloidin-positive apical microvilli on RPE cells was quantified using ImageJ software[22,24].

### Immunostaining and quantification of TFEB nuclear translocation using paraffin-embedded human retinal sections

Human control and AMD donor paraffin-embedded eye globes were subjected to dewaxing using three washes of fresh xylene and rehydration using a gradient of ethanol followed by antigen retrieval as explained previously[22,24]. The slides were then washed with PBS/0.1% Tween 20, twice for 10 min each, and then subjected to permeabilization and blocking steps as previously demonstrated[22,24]. The slides were incubated with TFEB primary antibody (A303-673A, Bethyl Laboratories) in primary antibody buffer (1% Tween 20 + 0.5% BSA in PBS) at a dilution of 1:100 and incubated in a moist chamber overnight at 4 °C. The slides were washed with wash buffer three times. A secondary antibody cocktail was prepared containing Anti-Rabbit Alexa fluor 647 (1:200) in secondary antibody buffer (0.1% Tween 20 + 0.5% BSA in PBS) also containing Hoechst (1:2000) and was added to each slide covering the sections completely and incubated in dark for 2 h at RT. Post-staining the slides were washed with 1× PBS, 5 times, for 15 min each and then mounted with a coverslip using a DAKO mounting media until imaging in a confocal microscope[22,24]. Slides were imaged using VS200 Olympus Slide Scanner with 40× objective (UPlanXApo NA 0.95) in spectral ranges: DAPI (ex. 378/52, em. 432/36), Cy3 (ex. 554/23, em. 595/31), Cy5 (ex. 635/18, em. 680/42) and Cy7 (ex. 735/28, em. 809/81) with ORCA Fusion BT digital CMOS camera. Cy7 channel was used to detect the RPE layer by thresholding followed by morphological operations. Individual nuclei were detected using Cellpose[56] generalist segmentation algorithm (model 'cyto') in the DAPI channel. TFEB-specific signal was calculated as a difference between Cy5 and Cy3 channels. Individual foci were detected using the Difference of Gaussian algorithm (scikit-image[57]). A nucleus was considered positive if more than two foci were detected within its area.

### Transmission electron microscopy

The WT and *Akt2* KI mouse eyes were immersed fixed in cold 2.5% glutaraldehyde (25% glutaraldehyde stock EM grade, Polysciences, Warrington, PA) and 2% paraformaldehyde (Fisher Scientific, Pittsburgh, PA) in 0.01 M PBS (sodium chloride, potassium chloride, sodium phosphate dibasic, potassium phosphate monobasic, Fisher Scientific, Pittsburgh, PA), pH 7.3. The sclera, choroid, and retina were separated and quartered after the optic nerve was removed[43]. The sample was rinsed 3× in PBS, post-fixed in 1% osmium tetroxide (Electron Microscopy Sciences) with 1% potassium ferricyanide (Fisher Scientific) for one hour, dehydrated by putting the samples in a series of ethanol (30–90%), and propylene oxide (Electron Microscopy Sciences). Next, the samples were properly oriented and embedded in Poly/Bed® 812 (Dodecenyl Succinic Anhydride, Nadic Methyl Anhydride, Poly/Bed 812 Resin and Dimethylaminomethyl, Polysciences). Thin (300 nm) cross sections of the area adjacent to the optic nerve were cut on a Leica Reichart Ultracut (Leica Microsystems), stained with 0.5% Toluidine Blue in 1% sodium borate (Toluidine Blue O and Sodium Borate, Fisher) and examined under the light microscope. Ultrathin sections of the same area (65 nm) were stained with 2% uranyl acetate (Electron Microscopy Sciences) and Reynold's lead citrate (Lead Nitrate, Sodium Citrate, and Sodium Hydroxide, Fisher) and examined on JEOL 1400 Flash transmission electron microscope with AMT Biosprint 12, 12 mp camera (Advanced Microscopy Techniques).

Lysosomal characteristics of WT and *Akt2* KI animals aged 3–15 months were compared (5–WT and 4 *Akt2* KI animals, with 2–10 images analyzed per animal). Prior to the analysis, images were anonymized and subsequently reviewed by an expert who manually annotated lysosomes. Calculated parameters (the total area and the number of lysosomes) were normalized to the area of the tissue layer in the TEM micrographs located between the outer segments (at the apical side) and the edge of the tissue (at the basal side), based on expert and blinded manual annotations. The area of individual lysosomes was determined using the regionprops function of Scikit-image Python package[57].

### RNAseq analysis and bioinformatics

The RNAseq analysis was performed from RPE cells harvested from WT and *Akt2* KI mice at 4 months of age as a paid service from Novogene Inc. The bioinformatics analysis was performed to identify differentially changing autophagy genes using previously published gene sets. Average expression values of the particular genes in each sample were then calculated by the function "AverageExpression" in the Seurat package[43].

### Chromatin immunoprecipitation

Chromatin Immunoprecipitation (ChIP) was performed as previously described[13]. RPE cells were harvested from WT and *Akt2* KI mice and ChIP was performed using a commercially available kit (17-295, EMD Millipore). The following primers for the Atg9b promoter and negative control were used Atg9B_CHIP_F1: CACATGCTAGAGCCCTCTGATA, Atg9B_CHIP_R1: GGCTCAAGAGCTATTGGGATT, Atg9B_CHIP_F2: GGC ATGTGGCCTTAAATCAT, Atg9B_CHIP_R2 GGCTCTAGCATGTGCAGA CA, Atg9B_CHIP_NEG_F: TGGCCATATCTGGGAGTCAA, Atg9B_CHIP_NE G_R: CCAGCCAGGCTAGTGTTCTC.

### Co-immunoprecipitation

ARPE19 cells were either transfected with EGFP-AKT2 or HA-SIRT5 or EGFP-AKT2 (K14A/R25E) (86593; 24483; 86595, Addgene) constructs using a Lipofectamine transfection kit (L3000008, Thermo Fisher)[21,22]. After 48 h the cells were lysed and a pulldown assay was performed from cell lysates using either anti-HA (88836, Thermo Fisher) antibody or anti-GFP (gtd-100, ChromoTek) magnetic beads followed by western blotting for respective proteins of interest.

### Human high-throughput protein–protein interaction

The evaluation of AKT2 binding partners was performed as a paid service from CDI NextGen Proteomics, Baltimore, MD, USA, as previously described[22]. Briefly, recombinant AKT2 was used for this study using the HuProtTM v3.1 human proteome array and the sample was placed on array plates at a concentration of 1 μg/mL. The results were analyzed using GenePix software as described previously and represented the significance of the probe binding signal difference from random noise (Z-Score). Only protein interactions with a Z-score above 6 were considered[22].

### Molecular modeling

Models of molecular structure for human RAC-beta serine/threonine-protein kinase (AKT2) and NAD-dependent protein deacylase sirtuin 5 (SIRT5) were obtained from the AlphaFold Protein Structure Database (ebi.ac.uk), models AF-P31751-F1-model_v4.pdb and AF-Q9NXA8-F1-model_v4.pdb, respectively. Molecular modeling of the possible interaction between AKT2 and SIRT5 was done using the 'movement' tool in UCSF CHIMERA, VER. 1.17.1.

### In vitro AKT2 activity measurement

ARPE19 cells were overexpressed with EGFP-AKT2 with or without HA-SIRT5 construct. After 48 h, the cells from all the experimental groups were lysed in CHAPS lysis buffer, followed by pulldown with anti-GFP

magnetic beads. The AKT2 complex was incubated with recombinant TFEB (TP760282, Origene) for 15 min at 37 °C in the presence of 10 nm ATP (A1852-1VL, Sigma Aldrich). The mixture was denatured using LDS sample buffer (NP0007, Invitrogen) containing β-marcaptoethanol and then SDS-PAGE was run. Phospho-protein staining was performed using a commercially available kit (P005A, ABP Biosciences), to evaluate the levels of TFEB phosphorylation. The phospho-protein band intensity in each experimental group was calculated by ImageJ software and a graphical representation was performed as the ratio of the phospho-protein to the total protein band intensity for each sample[43].

## Quantitative polymerase chain reaction (qPCR)
The expression of different genes was evaluated by qPCR as described previously[43], using a commercially available cDNA preparation kit (11754-050, Invitrogen) and Taqman probe-based qPCR master mix (4369016, Thermo Fisher). The commercially available Taqman probes (Thermo Fisher) used for this assay were; *Ctsd* (Mm00515586), *Ctsb* (Mm01310506), *Lamp1* (Mm00495262), and Atp6v0a1 (Mm00444210).

## Western blot
Western blotting was performed using previously published methods from our laboratory[22,24,43]. The cell and tissue lysates were prepared in 1× RIPA buffer (20–188, EMD Millipore) containing 0.1% of protease inhibitor cocktail (I3786, Sigma Aldrich) and 0.1% phosphatase inhibitor cocktail (P0044-5ML, Sigma Aldrich). Densitometry was performed to estimate the protein expression relative to the loading control (Actin) using ImageJ software (National Institute of Health)[22,24,43].

## Enzyme-linked immunosorbent assay (ELISA)
To evaluate the levels of IL-1β and HMGB1 in spent medium from WT and *Akt2* KI RPE explant cultures[22], as well as in *CFH* (Y/Y) and *CFH* (H/H) iPSC-derived RPE cell cultures, commercially available kits were purchased and the assays done following the manufacturer's protocol: IL-1β (Mouse: BMS6002; Human: BMS224INST, Thermo Fisher) and HMGB1 (Mouse: NBP2-62767; Human: NBP2-62766, Novus Biologicals).

## Estimation of cathepsin D and L activities
The estimation of cathepsin D (ab65302, Abcam) and L (ab65306, Abcam) activities from RPE lysates from WT or *Akt2* KI mice or from iPSC-derived RPE cells was performed by using commercially available kits.

## Generation of TFEB KO MEFs
*Tfeb* KO stable cell line was generated on the Trf2F/F; Rosa-CreER MEFs background[58]. The cells were infected with LentiCRISPR.v2 puro construct carrying the following guide sequences: Tfeb-3F: caccg-CAGCCCGATGCGTGACGCCA and Tfeb-3R: aaacCTGGCGTCACG CATCGGGCTG which targets exon 1 of mouse *Tfeb* gene. After infection, cells were treated with puromycin for selection and single clones were isolated to get the *Tfeb* KO MEF cell clones for further propagation.

## Seahorse
ARPE19 cells were plated at 40,000 cells per well on a Seahorse XF platform compatible 96-well plate pre-coated with poly-D-lysine (Sigma Aldrich, USA) and allowed to adhere and grow for 24 h[36]. The cells were transfected with either EGFP-AKT2 or EGFP-AKT2+HA-SIRT5 constructs or left untransfected and after 48 h were subjected to the Mitostress assay kit (Cat# 103015-100) from Agilent, USA[36].

## Hematoxylin-eosin staining
Eyes from *Cryba1*-floxed and trehalose or vehicle (water) treated *Cryba1* cKO mice were fixed in 2.5% glutaraldehyde followed by formalin, and then subjected to ethanol gradation and dehydrated followed by embedding in methyl methacrylate[22]. Sections (1 μm) were cut and stained with hematoxylin and eosin and observed under a light microscope as described previously. The quantification of drusen-like deposits has been described previously[22]. The evaluation of alterations in the ONL resulting from photoreceptor death and degeneration involved a quantitative analysis through the enumeration of nuclei in both central and peripheral regions of the retina. Hematoxylin-eosin (H&E) stained sections, comprising three sets with six sections each, were subjected to microscopic examination at a magnification of 40×. Image acquisition was performed, and subsequent analysis was conducted using MetaMorph software (MetaMorph Inc.). In the image analysis process, the RGB threshold was applied, and channels were separated to identify optimal resolution in the ONL, particularly within the green channel. To ensure precision and eliminate manual errors and counting bias, a systematic approach involved manually tracing a line exceeding 615 pixels along the ONL. This guided the software to generate a consistently sized rectangle in every image. Surface area measurements of nuclei were obtained by differentiating the threshold of the nuclei from the background. The average area of individual nuclei was determined to be approximately 115 pixels. Densely packed nuclei were identified by adhering to the pre-set threshold in the green channel. The cumulative surface area occupied by these nuclei was logged. The ratio of the total surface area to the area of each nucleus provided the total number of nuclei within the fixed area of each H&E panel.

## Fundus imaging
Fundus photographs were obtained from WT and *Akt2* KI mice using the Micron IV Laser Scanning Ophthalmoscope (Phoenix Research Lab, Inc) as previously described[21].

## Electroretinography
WT and *Akt2* KI mice at 4 and 10 months of age were dark adapted for 20 h and then were anesthetized with ketamine (50 mg/kg body weight) and xylazine (10 mg/kg body weight) and subjected to electroretinography to evaluate retinal function by estimating the scotopic a-, b- and c-wave responses using the Celeris Diagnosis System, USA[23]. Responses were measured at three different light intensities (0.01, 0.1, and 1 cd*s/m²)[23].

## Autophagy flux
Autophagy flux on RPE explants from WT and *Akt2* KI mice as well as *Cryba1* KO (±trehalose; 100 mM for 24 h) mice was performed as described previously[43]. Briefly, the explants were treated with Bafilomycin A1 (BafA1; 1 μm) or chloroquine (ChQ; 50 μM) or left untreated for 6 h[21,43]. Western blotting was performed to evaluate the levels of LC3-I and LC3-II. Autophagy flux was estimated by calculating the ratio of LC3-II in BafA1 or ChQ-treated cells with respect to untreated cells[43].

## Immunofluorescence analysis of autophagosome/autolysosome formation and lysosome/autophagosome fusion
To evaluate the levels of autophagosome and autolysosome formation upon AKT2 overexpression in RPE cells, the ARPE19 cells were transfected with either an AKT2-pcDNA construct (86623, Addgene) or left untreated for 48 h and then infected for 12 h with 10³ vg/ml of an Adenovirus-GFP-RFP-LC3 construct (Vector Biolabs, 2001)[43]. The cells were fixed with 2% PFA for 30 min at 4 °C and the nuclei were stained with Hoechst at 1 mg/100 ml dH₂O. Confocal images were acquired on the Zeiss LSM 710 platform (Switzerland). The data were analyzed using ImageJ software as previously described[43] and represented as puncta number for autophagosomes (yellow) and autolysosomes (red) in control and AKT2 overexpressing cells. Lysosome/autophagosome fusion was estimated in ARPE19 cells overexpressing only GFP-LC3

(24920, Addgene) or both GFP-LC3 and AKT2 constructs. After 48 h the cells were fixed (as mentioned above) and then immunostained with Lamp1 antibody (ab24170, Abcam), followed by secondary antibody staining (Goat Anti-rat Alexa fluor 455; 1:300) as explained previously[21,22]. Analysis of colocalization between LC3 (green) and Lamp1 (red) was performed using the JACoP plugin of ImageJ software to evaluate Pearson's co-efficient and the level of colocalization between the two proteins was estimated as previously described[43].

### Total internal reflection fluorescence (TIRF) microscopy

TIRF imaging was performed using an iLas2 Ring TIRF illuminator on a Nikon Ti inverted microscope equipped with a 1.49 N.A. 100× Nikon TIRF objective and 1.5× tube lens with 0.07 effective pixel size[41]. Images were acquired using a Photometrics Prime 95B sCMOS camera. Images of the ARPE19 cells with or without AKT2 overexpression for 48 h followed by infection with $10^3$ vg/ml of Adenovirus-GFP-RFP-LC3 construct for 12 h were collected in serum-free starvation medium for 2 h[43]. Images were analyzed using NIS Elements spot detection in order to identify punctate structures at the membrane. The number of spots was counted over time and each field of view (FOV) was normalized to the average number of spots in the FOV to compare relative changes between all analyzed FOVs. The quantification was performed using Graph Pad Prism 8 software to ascertain the difference in between the two groups across all the time points using linear regression plots.

### Statistical analysis

All analyses were performed using the GraphPad 8.0 software and Microsoft Excel. Student's t-test (experiments with two groups) and one-way ANOVA followed by Tukey post-hoc test (experiments with more than two groups) were used to measure the statistical differences between groups[22,43]. The significance was set at $P < 0.05$. The analyses were performed on at least three biological replicates (n)[22,43]. All the values are presented as mean ± standard deviation (SD)[22,43].

### Reporting summary

Further information on research design is available in the Nature Portfolio Reporting Summary linked to this article.

## Data availability

All data generated or analyzed during this study are included in this published article (and its Supplementary Information files). The RNA-seq data for the genes depicted in Fig. 4c has been deposited to GEO NCBI with Accesion number: GSE269923. Fig. 6j was created with BioRender.com. Source data are provided with this paper.

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

## Acknowledgements

This work was supported by NIH R01 EY031594 (to D.S. and J.T.H.), NIH R01 EY032516 (to D.S.), NIH R01 EY028554 (to D.A.F. and J.R.D.), Edward N. and Della L. Thome Memorial Foundation Awards Program in Age-Related Macular Degeneration Research (to D.S.), BrightFocus Foundation Postdoctoral Fellowship on Macular Degeneration (to S.G.), start-up funds to D.S. from Ophthalmology, University of Pittsburgh, the Jennifer Salvitti Davis, M.D. Chair Professorship in Ophthalmology (D.S.), The Robert Bond Welch Professorship (J.T.H.), P30 core award EY08098 from the National Eye Institute, NIH (to the University of Pittsburgh Department of Ophthalmology), and unrestricted funds from The Research to Prevent Blindness Inc., NY (to the University of Pittsburgh Department of Ophthalmology and the Wilmer Eye Institute) and the Academy of Finland grant 333302 (to K.K.). The authors would also like to acknowledge the bioinformatics and imaging core facilities within the Department of Pediatrics, University of Pittsburgh School of Medicine, and Kaarniranta AMD lab staff for *PGC-1α* KO mice generation and sample isolation. The authors would like to thank the staff of the Center for Biologic Imaging, University of Pittsburgh for their assistance with processing TEM and live cell imaging samples. This project has also been made possible in part by grant number 2020-225716 from the Chan Zuckerberg Initiative DAF, an advised fund of the Silicon Valley Community Foundation.

## Author contributions

D.S. designed and conceptualized the study. J.T.H. provided input about the genetic aspects of the study. S.G., S.B., V.K., M.N., and M.Y. performed the experiments. R.S., D.B. J.M., J.R., and K.B. generated the iPSC-derived RPE cells lines (from non-AMD donors) and the *CFH* null and isogenic control cells (iRPE). P.S., H.M.A., J.R.D., Z.G., and D.A.F. generated iPSC-derived RPE cell lines from human AMD donors. R.S., D.B. J.M., and J.R. performed APOE and BODIPY staining on iPSC-derived RPE cells. K.M.K. and D.B.S. performed quantification of TEM and immunofluorescence data. C.T.W. and S.C.W. performed and analyzed TIRF experiment results. J.F. performed the TEM imaging. C.Y.W. and T.F. generated the *Tfeb* KO and *Tfeb*-flox MEF line. Y.S. performed the computer modeling. D.R. performed bioinformatics analysis. E.S.G. and K.K. provided the *Sirt5* KO and *PGC-1α* KO mice. S.G., D.S., R.P., S.H., A.S., M.F.B., J.A.S., J.T.H., and J.S.Z. analyzed the data. S.G., D.S., S.H., A.S., J.S.Z., and J.T.H. wrote the paper.

## Competing interests

The authors declare no competing interests.

## Additional information

[1]Department of Ophthalmology, University of Pittsburgh School of Medicine, Pittsburgh, PA, USA. [2]Ocular and Stem Cell Translational Research Section, National Eye Institute, National Institutes of Health, Bethesda, MD, USA. [3]Department of Cell Biology, Center for Biologic Imaging, University of Pittsburgh School of Medicine, Pittsburgh, PA, USA. [4]Aging Institute, University of Pittsburgh School of Medicine, Pittsburgh, PA, USA. [5]Doheny Eye Institute, Pasadena, CA, USA. [6]Stem Cell Institute, University of Minnesota, Minneapolis, MN, USA. [7]Department of Pediatrics, University of Pittsburgh School of Medicine, Pittsburgh, PA, USA. [8]Institut De La Vision, INSERM, CNRS, Sorbonne Université, Paris, France. [9]Cell Biology and Physiology Center, National Heart, Lung, and Blood Institute, National Institutes of Health, Bethesda, MD, USA. [10]Wilmer Eye Institute, The Johns Hopkins University School of Medicine, Baltimore, MD, USA. [11]Protein Biochemistry & Molecular Modeling Group, National Eye Institute, National Institutes of Health, Bethesda, MD, USA. [12]Department of Ophthalmology, University of California Los Angeles, Los Angeles, CA, USA. [13]Department of Ophthalmology, University of Colorado Anschutz Medical Campus, Aurora, Colorado, USA. [14]Department of Ophthalmology, University of Eastern Finland and Kuopio University Hospital, Kuopio, Finland. [15]Department of Molecular Genetics, University of Lodz, Lodz, Poland. ✉e-mail: kapil.bharti@nih.gov; jthanda@jhmi.edu; Debasish@pitt.edu

