## [Peer Review File · Nature Communications]

The AKT2/SIRT5/TFEB pathway as a potential therapeutic target in non-neovascular AMDREVIEWER COMMENTS

Reviewer #1 (Remarks to the Author):

Atrophic age-related macular degeneration (AMD) is a multi-factorial disease that is accompanied with degeneration of the aging retinal-pigmented epithelium (RPE). One of the main processes that maintain the viability of RPE cells is lysosomal-mediated waste clearance/autophagy. This is regulated by the transcription factor TFEB and TFE3 as well as the mTOR-AKT2 signaling pathway. Disruption of these processes including the signaling pathways involved contributes mechanistically to AMD development. There are multiple risk factors for the disease, including genetic factors such as the complement factor H (CFH)-Y402H. However, the molecular mechanism involved in the influence of this and other risk alleles and their downstream effects on lysosomal function, is not fully elucidated. The present study was undertaken to determine the role of the AKT2-dependent effect on lysosomal function and autophagy in the RPE, leading to the disease pathology. Specifically, the authors found that RPE with CFH Y402H mutation have an increased phosphorylation of AKT-2 that leads to the formation of an AKT-2- SIRT-5 complex. This causes in turn a reduction in AKT-2 malonylation that reduces its kinase activity and attenuates the TFBE-dependent lysosomal activation and function.

Overall, this is a well written study that presents interesting findings, particularly the link between AKT-2 and AMD. However, the mechanism that governs this phenomenon is poorly proven and some of conclusions lack suitable evidence. Also, some experiments lack appropriate controls. These, as well as a few other comments listed below should be addressed to justify publication in Nature Communication. These are as follows:

Comments

- 1) In Figure S1a, it is important to show that the knock-in of AKT-2 has been successful (i.e AKT-2 is overexpressed compared to the WT) and whether the overexpressed AKT-2 is phosphorylated or not.
- 2) In connection to it, it would be informative to demonstrate the status of endogenous AKT2 and pAKT2 from samples of patients who suffer from AMD as compared to healthy individuals.
- 3) Figure 1A: the photos are of high quality and the phenotypic differences are significantly evident. I kindly ask the authors to also add quantification of the number of dysmorphic cells when comparing the WT to the AKT-2 KI mouse.
- 4) Figure 1E: Please provide information on the efficiency of conditional AKT-2 knockout, by western blot of the KO mice compared to WT).
- 5) Figure S2C: This experiment is not convincing as the input of 14-3-3 is not equal in the two samples. In addition, it would be important to show the level of AKT and pAKT in the samples.

- 6) Figure 2D: Please add blots showing AKT-2 and p-AKT to prove the claim.
- 7) Figure S4 C, D: The way the nuclei were compared between the “control” and the “AMD donors” is not well explained. Are the changes significant? Also, when relating to nuclear translocation it is advised to show the same result also by using biochemical fractionation.
- 8) Figure 3b: The results are not convincing. A control showing the specificity of IP or CoIP should be added (using an IgG antibody). It would be also informative to show pAKT2 both in the cell extract or in the CoIP.
- 9) Figure 3C: These results are not convincing. Not only the difference in the phospho protein is very minor (lane 6), the “total protein” in the HA- SIRT and AKT-2 coinfection (lane 6) is markedly elevated which is probably the reason for the elevation of the phospho-protein. It is recommended also to use controls such as AKT inhibitors to further validate this issue.
- 10) Figure 3D: The fact that AKT2 activity is regulated by malonylation of interacting proteins has not been shown, and therefore, the finding described here may be very interesting to the AKT2 field. However, the results here are not convincing, as the amount of AKT2 in the third lane of the lower panel is much lower than in the second lane, which correlate to the reduction in K-mal (upper panel). In addition, there is no results demonstrating that the malonylation affects at all the activity of AKT-2.
- 11) PCG-1a is known to be regulated by AKT-2. Here the authors show that PCG-1a can also regulate AKT2, which is important for the field. However, this point is not rigorous enough and should be further validated.
- 12) Figure 4D: The western blot of “ULK1” is of poor quality and seems to be manipulated. Please provide a new one.
- 13) Figure 5: It is important to show again blot of AKT-2 and pAKT and maybe use an AKT inhibitor as another way to validate the existence of an AKT- SIRT-5 – TFEB axis. Also, if the AAV2-TFEB-S467A induces constitutive nuclear localization of TFEB, this should be shown by staining. Again, it would be better to also add fractionation to further validate the claim.
- 14) What kinase phosphorylates AKT-2 in the cells from AMD patients and is it the T308 or S473 site? This can be easily shown using the appropriate phosphor antibodies and the corresponding kinase inhibitors.
- 15) The authors should better explain the specificity of AKT2 in this system. Can the results be reproduced with AKT1/3? What are the mechanisms that contribute to the specificity.
- 16) Although the manuscript is generally well-written, there are a few minor inaccuracies throughout the article. For example, the term “increased AKT” is problematic as it may refer to either activity or expression. In addition, in Figure S1E it is very difficult to see the red color.

Reviewer #2 (Remarks to the Author):

Main Comments

Noteworthy results

The manuscript by Gosh et al. presents data that potentially offer a mechanism that is involved in the pathology of age-related macular degeneration (AMD). The group show data from a AKT2-KI transgenic mouse model indicating an AMD like phenotype in aged mice compared to their WT littermates. Furthermore, the group used iPS-generated RPE cells (iPS-RPE) from human donors that carry an AMD risk-associated polymorphism in the gene of the complement factor H (CFH-Y402H) that show lysosomal dysfunction potentially associated with higher AKT2 activity.

Significance in the field

The data could be of interest for a better understanding of the pathology of AMD. However, I think that the authors did not succeed to show the relevance of an overactivity by AKT2 in the pathogenesis in AMD. This has the following reasons:

1. The authors state: "It was previously reported that AKT2 is upregulated in dysmorphic macular RPE cells from AMD donor globes 13,14". Paper 13 shows for human retinas with AMD features (dysmorphic RPE?) increased LCN-2 expression, a transcription factor that drives inflammation; and this is what the paper is about. LCN-2 might be activated by AKT2 and NFkappaB but these are not the only pathways. Paper 14 reported a role of miRNA184 in RPE differentiation; a micro-RNA that targets AKT2 expression among other RNAs. The group showed in an eye from one donor increased AKT2 mRNA levels compared to one age-matched control. The authors concluded a dysfunctional RPE, but I do not know dysfunctional in which way. The authors did not show decreased levels of miRNA184. It is rather so that RPE cells are dedifferentiated in AMD. As only one donor is analyzed, we have clue whether this is a regular observation in AMD.
2. AKT2-KI mouse model shows dystrophic RPE cells with age that is more frequent than in the WT controls. Age-dependent changes indicate faster progression of a degenerative phenotype compared to WT animals. This is very interesting but an AMD-like phenotype by overactivity of AKT2 in mouse does not prove that AKT2 is causing AMD in humans. It can be that AKT2 mechanisms have secondary effects due to ongoing degeneration. Furthermore, AKT2 is a highly important kinase in photoreceptors. Li et al. (2007 J Neurosci) showed that AKT2 acts in a non-redundant manner together with AKT1 in the protection of photoreceptors against light-damage. Thus, an impact of overactive AKT2 in photoreceptors should not be overseen when claiming that AKT2-KI is a RPE-based model for AMD.
3. The link for human AMD should be given by the usage of iPS-RPE cells from a donor carrying the risk allele CFH-Y402H. The authors claim this to be a "disease in the dish model" for AMD because AMD starts in the RPE. This argument is questionable because other observations show that AMD might also start in the photoreceptors or in the choroid. That the iPS-RPE cell culture represents a model for AMD is supported by a paper (reference 20) that has not been replicated so far. The paper that is cited here shows in one iPS-RPE cell line with CFH-Y402-H that somehow complement is

dysregulated leading to increased levels of terminal complement complex (C5b-9) in lysosomes which in turn leads to lysosomal dysfunction. The cited paper leaves many questions open and does not provide a valid link between CFH polymorphism and lysosomal dysfunction (C5b-9 detection was performed with anti-C9 antibodies; can be that it is C9 and not C5b-9 in the lysosomes).

4. The authors in this study, did not try to at least recapitulate the data from reference 20 to support at least the relevance of iPS-RPE with CFH-Y402H so that the lysosomal dysfunction is comparable and a general feature in in this model. However, the authors provide data how AKT2 might lead to dysfunctional lysosomes and claim that they provide the link between CHF polymorphism and lysosomal dysfunction. It remains unclear how CFH-Y402H activated AKT2. The authors state that polymorphic CFH accumulates in lysosomes as a mechanism. The focus onto CFH-Y402H is important because the polymorphism that is the most relevant difference between the used cell line models. There is no evidence for a mechanism how CFH (not C5b-9) lysosomal accumulation activated AKT2. Thus, there is a missing link between the mouse model and the human condition.

5. The authors neglect a large body of evidence about the role of CFH-Y402H in AMD. In the light of this evidence their own data lose significance. This body of evidence clearly demonstrates that the CFH-Y402H role in the AMD pathology lies in the extracellular control of the complement system. Among these observations are those that indicate increased levels of C5b-9 and other complement factors in the plasma/serum from AMD patients. Locally, accumulation of complement components in the outer retina (Drusen, Bruch's membrane etc.) appeared characteristic (excellently reviewed by Clark and Bishop (2018 Semin. Immunopathol) or Toomey et al. (2018 ProgRetinEyeRes). The complement activity regulation under CFH-Y402H shows weak control of C3 convertase and thrombospondin activity. For that reason, the link between CFH-Y402H, lysosomal dysfunction and AKT2 must be stronger demonstrated.

Support of the conclusions by the data

A couple of data weaken the conclusions:

1. The AKT2-KI phenotype: I am somewhat puzzled by the data. At month 12 the WT mice show a dense rhodopsin staining in the IS/OS. In the AKT2 mouse of the same age there apparently no rhodopsin present. The thickness of IS/OS seems to be normal. Thus, there is no photoreceptor degeneration but loss of rhodopsin. Therefore, AKT-2 overactivity has a severe impact onto photoreceptors. In the light of this observation, the AKT2 overactivity in the RPE loses its relevance. Indeed, AKT2 is highly active in photoreceptors (Li et al. 2007 J Neurosci).

2. Why there are data from the mouse in a mix of 12 months and 15 months old animals? Is there a bias for selected results?

3. Is there a way to show the complete western-blot and not only single bands?

4. Conditional knockout: It seems that the Cre-Lox system is used. Cre expression in the RPE has been found to be toxic for the RPE; applies especially for the Cryba1 model.

5. Cathepsin D/L activities in human AMD retina: this is an important piece of the data. However, it is to mention that three controls and three AMD retinas were analyzed. The data show that the

activities of cathepsin D and L were decreased in the AMD probes. This can have other reasons than reflecting a direct disease mechanism. RPE cells in geographic atrophy degenerate and display a reduced metabolism with decreased protein expression and phosphorylation activities. Thus, cathepsin activity reduction might be effect of the degenerative processes and not cause. Why did the authors not measure AKT2?

6. IP experiments: When IP was performed with an anti-GFP antibody AKT2 can be precipitated when using the GFP-AKT2 plasmid transfection but becomes weaker in the GFP-AKT2/SIRT5-HA double transfection. Why? Or is this just an observation in this blot?

7. CFH KO mouse: this model needs to be taken with care. Due to the lack of CFH the mice are completely lacking C3. It is hard to differentiate between the lack of CFH and C3.

8. In their previous paper, the authors showed that AKT2 KO in the mouse leads to compensatory upregulation of AKT1 that in turn participates the pathology of diabetic retinopathy. Is AKT1 downregulated in AKT2 KI mice? Can it be that the pathology lies on a AKT1-dependent pathway?

9. iPS-RPE cells: Among the cell lines this model is widely accepted as being a very relevant model. However, also these cells bear strong quality impairments. The paper lacks a validation of the cell line, that is also several times passaged. First, the authors used only one cell line from one donor. It is known that even different cell lines from the same donor display different expression activities of RPE markers (see as an example Marmorstein et al. 2018 Sci Rep). We cannot exclude that the differences between the CHF-Y402H cell line and the CFH-WT cell line are due to variances in gene expression of lysosomal pathways or AKT2 signaling. iPS-RPE react very sensitive on passaging, another inducer of variability. The investigation of one cell line from one donor and one from a control donor is not enough for generalizations. And a very important question, when it comes to lysosomes: are the iPS-RPE used in this study pigmented?

10. AKT2 overactivity in mouse: The authors present a large body of data that precisely describe the chain of events that lead to impaired lysosomal function, and in turn to increased secreted autophagy by RPE cells. Indeed, Figure 1 shows kind of basal lamina deposits in the AKT2-KI. However, how can the authors exclude basal laminar deposits in the 15 months old WT? Furthermore, At the end of the results part the authors claim that Trehalose treatment abolishes the basal laminar deposits. This is not shown.

11. Most of the Trehalose intervention studies were performed with the Cryba1 mouse model. Especially the most relevant data on retinal structure. However, this does not proof the AKT2 hypothesis. Why were Trehalose experiments not done with AKT2-KI?

12. There is a spidergram in the last figure that shows retinal degeneration in the CRyba1 model. This not shown for AKT2. However, given the retinal sagittal sections, I can not recapitulate the IS/OS ratios. Why not counting the number of rows of nuclei in the ONL?

Gender

Epidemiology shows differences between male and females in the incidence and risk for AMD. Gender differences are of importance.

Human tissue: No gender data of the donors used for analysis of human tissue; only three with AMD and three age-matched controls.

iPS-RPE: no data whether these are male or female cells.

Mouse experiments: No indications about the gender of mice used in the study.

Reviewer #3 (Remarks to the Author):

This is an interesting study describing the potential role of AK2/SIRT5/TFEB pathway in lysosomal dysfunction in AMD. The strength of the study is the use of different models, namely the Akt2 knock-in mouse, AMD patient Y402H RPE cells, AMD donor eyes and ARPE19 cells to demonstrate accumulation of AK2, which impairs TFEN/TFE3 lysosomal function via inhibition of PGC-1 α and SIRT5. While the work has been carefully conducted, there are details missing in both text and figure legends, which should be added at the revision stage. Further, the data has not been widely discussed in the context of other publications and thus there is scope to balance their findings against previous work that have not only shown lysosomal dysfunction in AMD RPE cells, but also provided mechanistic insights. Below there are some questions/comments that will strengthen the manuscript:

1. How many iPSC-RPE and isogenic controls were used for the study? Apart from the Y402H polymorphism, were there any genotype changes in other complement genes (e.g. C3, C5)? If yes, were there any changes in AK2 expression between iPSC-RPE derived from AMD patients with only Y402H and Y402H+ other complement risk alleles?
2. The authors show changes in pTFE-3 and pTFEB expression in AMD donors: this work needs to be validated in the Y402H-RPE cells and in the Akt2 KI RPE cells.
3. Since TFEB is a master regulator of lysosomal biogenesis and the authors are describing lysosomal dysfunction, one would have expected changes in lysosome number and/or volume/surface area. None of these parameters have been described. These data can be easily obtained from the existing TEM data.
4. In figures 5 and 6 changes in expression of CTSL and CTSD have been described in response to various treatment, however, to reach the conclusion that lysosomal function has been rescued, enzyme activities need to be carried out to exclude the potential scenario of these proteins being secreted in the media.
5. Figure 1c: higher magnification images are needed to substantiate the presence of Blam deposits.
6. Seahorse assays have only been performed in ARPE19 cells overexpression Akt2 or Akt2+Sirt5. To validate these findings, it would be necessary to repeat the seahorse assays in the Y402H and isogenic control derived RPE cells.

7. Flores-Bellver described recently increased exocytosis in RPE cells treated with AMD stressors: the authors have made no attempt to discuss their findings in the context of these published data.

March 26, 2024

Re: **Manuscript Number: NCOMMS-23-38549-T, titled “*The AKT2/SIRT5/TFEB pathway as a potential therapeutic target in atrophic AMD*”**

Please find our detailed responses to the reviewers' comments highlighted in blue below.

REVIEWER COMMENTS

Reviewer #1 (Remarks to the Author):

Atrophic age-related macular degeneration (AMD) is a multi-factorial disease that is accompanied with degeneration of the aging retinal-pigmented epithelium (RPE). One of the main processes that maintain the viability of RPE cells is lysosomal-mediated waste clearance/autophagy. This is regulated by the transcription factor TFEB and TFE3 as well as the mTOR-AKT2 signaling pathway. Disruption of these processes including the signaling pathways involved contributes mechanistically to AMD development. There are multiple risk factors for the disease, including genetic factors such as the complement factor H (CFH)-Y402H. However, the molecular mechanism involved in the influence of this and other risk alleles and their downstream effects on lysosomal function, is not fully elucidated. The present study was undertaken to determine the role of the AKT2-dependent effect on lysosomal function and autophagy in the RPE, leading to the disease pathology. Specifically, the authors found that RPE with CFH Y402H mutation have an increased phosphorylation of AKT-2 that leads to the formation of an AKT-2- SIRT-5 complex. This causes in turn a reduction in AKT-2 malonylation that reduces its kinase activity and attenuates the TFBE-dependent lysosomal activation and function.

Overall, this is a well written study that presents interesting findings, particularly the link between AKT-2 and AMD.

The authors thank the reviewer for acknowledging the significance of our study in elucidating the novel association between AKT2 and AMD.

However, the mechanism that governs this phenomenon is poorly proven and some of conclusions lack suitable evidence. Also, some experiments lack appropriate controls. These, as well as a few other comments listed below should be addressed to justify publication in Nature Communication.

These are as follows:

Comments

1) In Figure S1a, it is important to show that the knock-in of AKT-2 has been successful (i.e AKT-2 is overexpressed compared to the WT) and whether the overexpressed AKT-2 is phosphorylated or not.

Agreed. The specificity and efficiency of AKT2 knockin in RPE cells and the expression of total AKT2 and p-AKT2 (S474) has been shown previously from our lab (PMID: 36229454). As this information was previously published, we have referenced this prior publication in the text (please see Page 6, lines 125-127 of the revised submission). We thank the reviewer for raising this important point.

2) In connection to it, it would be informative to demonstrate the status of endogenous AKT2 and pAKT2 from samples of patients who suffer from AMD as compared to healthy individuals.

Agreed. In response to the reviewer's suggestion, the authors have incorporated additional results in the revised manuscript demonstrating that the ratio of phosphorylated AKT2 (pAKT2) to total AKT2 is elevated in RPE cells obtained from human donors with AMD compared to those from healthy individuals (please see Figure 2d of the revised manuscript). Thank you for this very helpful suggestion which we believe strengthens the clinical implications of our findings.

3) Figure 1A: the photos are of high quality and the phenotypic differences are significantly evident. I kindly ask the authors to also add quantification of the number of dysmorphic cells when comparing the WT to the AKT-2 KI mouse.

Agreed. The authors have added quantification of dysmorphic cells (that differ from the normal cobblestone like morphology) in the revised manuscript (please see Figure 1a and Supplementary Figure 1c).

4) Figure 1E: Please provide information on the efficiency of conditional AKT-2 knockout, by western blot of the KO mice compared to WT).

Agreed. The efficiency of the *Akt2* conditional knockout has been previously demonstrated by our group (PMID: 36229454). As this information was previously published, in the text we have referenced this prior publication (please see Page 21, Line 493).

5) Figure S2C: This experiment is not convincing as the input of 14-3-3 is not equal in the two samples. In addition, it would be important to show the level of AKT and pAKT in the samples.

Agreed. The co-immunoprecipitation experiment was repeated as per the reviewer's suggestion, ensuring the inclusion of appropriate input controls. Additionally, we have provided data

on the level of p-AKT2 (active form of the kinase) in all the samples (please see Supplementary Figure 3c). Thank you for bringing this to our attention.

6)Figure 2D: Please add blots showing AKT-2 and p-AKT to prove the claim.

Agreed. The authors have incorporated western blot results, as suggested by the reviewer, for pAKT2 and total AKT2. The results show that the pAKT2/AKT2 ratio is increased in human AMD donor RPE lysates as illustrated in Figure 2d of the revised manuscript.

7)Figure S4 C, D: The way the nuclei were compared between the “control” and the “AMD donors” is not well explained. Are the changes significant?

The quantification of individual nuclei was conducted utilizing a pre-trained neural network (Cellpose), complemented by the application of the Difference of Gaussians algorithm (these details are provided in the revised Methods section). Despite encountering variability in TFEB expression across samples, rendering the data statistically insignificant, noteworthy trends emerged. Particularly, a noticeable decrease in TFEB nuclear localization was observed among AMD donors compared to age-matched controls. This is now made clear in the text. Thank you for bringing this to our attention.

Also, when relating to nuclear translocation it is advised to show the same result also by using biochemical fractionation.

Agreed. The reviewer raises an important point and we agree with the reviewer’s suggestion. We tried this approach but measuring nuclear translocation via fractionation from RPE lysates necessitated a significant amount of human donor tissue, which was challenging to obtain due to the limited availability of dry AMD donor samples. We were therefore unable to detect expression of known nuclear proteins (e.g., histone 3) using this approach. We therefore instead conducted western blot analysis using whole cell RPE lysates to assess the nuclear status of phospho-TFEB/TFE3 proteins (see Figure 2d), following established protocols (PMID: 30120233, PMID: 24448649).

8)Figure 3b: The results are not convincing. A control showing the specificity of IP or CoIP should be added (using an IgG antibody).

Agreed. In response to the reviewer's feedback, we have improved the specificity validation of the CoIP method. This was achieved by employing GFP-AKT2 and HA-SIRT5 plasmids for co-transfection in ARPE19 cells. Subsequently, CoIP was conducted using HA-specific antibody and IgG, followed by a western blot analysis of AKT2 and HA (Figure 3b).

It would be also informative to show pAKT2 both in the cell extract or in the CoIP.

Agreed. The authors have provided the levels of AKT2 in both the cell extract and the CoIP fractions, as illustrated in Figure 3b.

9)Figure 3C: These results are not convincing. Not only the difference in the phospho protein is very minor (lane 6), the “total protein” in the HA- SIRT and AKT-2 coinfection (lane 6) is markedly elevated which is probably the reason for the elevation of the phospho-protein. It is recommended also to use controls such as AKT inhibitors to further validate this issue.

We apologize for the confusion. As pointed out by the reviewer, lane 6 in the figure does not represent any co-transfection or IP. Lanes 5-8 correspond to recombinant protein samples, namely TFEB (lane 5), TFEB+AKT2 (lane 6), TFEB+SIRT5 (lane 7), and TFEB+AKT2+SIRT5 (lane 8); this has been made clear in the revised figure legend. In response to the reviewer's suggestion, we have

also incorporated a specific AKT2 inhibitor (CCT128930; PMID: 31552301) to further validate our findings. Regarding concerns raised by the reviewer with the data in lane 6, the recombinant AKT2 and TFEB were directly added to the tube rather than being equally loaded IP complexes from cell lysates (as was performed in lanes 1-4). Consequently, upon gel electrophoresis, we observe a higher amount of total protein (manifested as a band around 55-60 kDa) compared to other lanes, as would be expected. Similarly, lanes 6-8 exhibit a lower band for SIRT5 in addition to the 55-60 kDa band.

0) Figure 3D: The fact that AKT2 activity is regulated by malonylation of interacting proteins has not been shown, and therefore, the finding described here may be very interesting to the AKT2 field. However, the results here are not convincing, as the amount of AKT2 in the third lane of the lower panel is much lower than in the second lane, which correlate to the reduction in K-mal (upper panel). In addition, there is no results demonstrating that the malonylation affects at all the activity of AKT-2.

We thank the reviewer for acknowledging the novelty and potential significance of our finding regarding the regulation of AKT2. The reviewer raises an important point that there are no results demonstrating that malonylation affects AKT2 activity. To prove this, in-depth genetic inhibition studies must be performed which are beyond the scope of the current manuscript. Therefore, we have omitted the malonylation results from the revised manuscript for better readership. However, the authors have included additional data supporting the capacity of SIRT5 to regulate AKT2-dependent processes: (i) TFEB phosphorylation (indicative of nuclear translocation) in ARPE19 cells overexpressing both SIRT5 and AKT2 (Figure 3d) and (ii) *in vitro* TFEB phosphorylation (Figure 3c). In addition, we also show that loss of SIRT5 can lead to the activation of AKT2 phosphorylation in RPE cells (Figure 3g). These findings collectively provide compelling evidence that AKT2 activity is regulated by SIRT5. We thank the reviewer for highlighting this observation and for the additional suggested experiments which we believe further strengthen our manuscript.

1) PGC-1a is known to be regulated by AKT-2. Here the authors show that PGC-1a can also regulate AKT2, which is important for the field. However, this point is not rigorous enough and should be further validated.

We thank the reviewer for raising this point. Previous research supports a direct inhibitory effect of Akt2 on PGC-1a (PMID: 17554339). Our study demonstrates a novel coregulatory relationship between AKT2 and SIRT5 mediated through PGC-1a. While the direct regulation of AKT2 by PGC-1a remains unexplored in this manuscript (and the existing literature), our findings shed light on an intricate interplay between these two molecules. In Figures 3e and f, we demonstrate reduced levels of PGC-1a and SIRT5 in Akt2 KI RPE cells, accompanied by increased AKT2 phosphorylation and decreased SIRT5 levels in PGC-1a KO RPE cells. These observations highlight mutual regulation between AKT2/SIRT5 and PGC-1a.

We appreciate the insightful observation of the reviewer regarding the regulatory influence of AKT2 on PGC-1a. We have revised the text to present our findings within the broader framework of existing knowledge on the regulation of AKT2 by PGC-1a. Our findings unveil the multifaceted interplay among AKT2, SIRT5, and PGC-1a, and offer new insights into their coordinated regulation with potential implications for cellular homeostasis and disease pathogenesis.

2) Figure 4D: The western blot of "ULK1" is of poor quality and seems to be manipulated. Please provide a new one.

Agreed. The authors apologize for the poor quality of the ULK1 blot. There was no data manipulation on this blot or any of the blots/experiments presented in this study. We have replaced

the ULK1 blot with another representative one in the revised manuscript in which the image is of higher quality (please see Figure 4d).

3) Figure 5: It is important to show again blot of AKT-2 and pAKT and maybe use an AKT inhibitor as another way to validate the existence of an AKT- SIRT-5 – TFEB axis. Also, if the AAV2-TFEB-S467A induces constitutive nuclear localization of TFEB, this should be shown by staining. Again, it would be better to also add fractionation to further validate the claim.

Agreed. The authors have validated the AKT2/SIRT5/TFEB axis by CFH (H/H)+CCHS (to simulate peripheral complement activation as seen in AMD) iPSC-derived RPE cell with a specific AKT2 inhibitor and found that LAMP1 and Cathepsin D levels were significantly rescued upon AKT2 inhibitor treatment (Figure 5f). In response to the reviewer's suggestion, the authors have also incorporated additional immunofluorescence data demonstrating that AAV2-TFEB S467A infection promotes TFEB nuclear translocation in *Akt2* KI RPE cells (please see Supplementary Figure 12). Collectively, these findings further suggest that AKT2 plays a critical role in modulating TFEB signaling in these cells. We thank the reviewer for these helpful suggestions which we believe further strengthen our manuscript.

4) What kinase phosphorylates AKT-2 in the cells from AMD patients and is it the T308 or S473 site? This can be easily shown using the appropriate phosphor antibodies and the corresponding kinase inhibitors.

In most cells, the PI3 kinase has been identified as the kinase that phosphorylates AKT2 (PMID: 28157504). We employed an antibody to assess the level of AKT2 S474 phosphorylation, which is required for AKT2 to achieve maximal kinase activity (PMID: 31548312, PMID: 28157504). Detailed antibody information is provided in the Methods section. We have clarified this in the revised text. We thank the reviewer for raising this point.

5) The authors should better explain the specificity of AKT2 in this system. Can the results be reproduced with AKT1/3? What are the mechanisms that contribute to the specificity.

In this study, our focus was on Akt2 based on the prior observation of increased AKT2 expression/activation in the macular RPE of human AMD donor eyes (PMID: 27418134) To this end, we used an RPE-specific *Akt2* KI, as described previously (PMID: 36229454). Evidence of AKT2 activation specificity in RPE (as opposed to neurosensory retina) cells in these mice is demonstrated in Supplementary Figure 1a. Our prior study demonstrated that Akt1 levels remain unchanged in RPE cells of *Akt2* KI mice (PMID: 36229454), thereby reinforcing that our observations using these animals are due to the increased expression of Akt2 rather than secondary changes in the expression of Akt1. Nonetheless, the reviewer raises an interesting question. We do not specifically examine the contribution of Akt1 or Akt3 in this study, which would require comprehensive experiments, and several additional tissue-specific KI/KO mice. We believe that these studies, while of potential interest, are beyond the scope of the present study.

6) Although the manuscript is generally well-written, there are a few minor inaccuracies throughout the article. For example, the term "increased AKT" is problematic as it may refer to either activity or expression. In addition, in Figure S1E it is very difficult to see the red color.

Agreed. We apologize for the confusion. In response to the reviewer's suggestion, we have carefully reviewed the manuscript for inaccuracies and addressed them in the revised text. Specifically, we have corrected the reference to "increased AKT2" and clarified whether it pertains to activity or expression. Additionally, we have improved the visualization of Supplementary Figure 1e for better clarity.

Reviewer #2 (Remarks to the Author):

Main Comments

Noteworthy results

The manuscript by Gosh et al. presents data that potentially offer a mechanism that is involved in the pathology of age-related macular degeneration (AMD). The group show data from a AKT2-KI transgenic mouse model indicating an AMD like phenotype in aged mice compared to their WT littermates. Furthermore, the group used iPS-generated RPE cells (iPS-RPE) from human donors that carry an AMD risk-associated polymorphism in the gene of the complement factor H (CFH-Y402H) that show lysosomal dysfunction potentially associated with higher AKT2 activity.

Significance in the field

The *data could be of interest for a better understanding of the pathology of AMD*. However, I think that the authors did not succeed to show the relevance of an overactivity by AKT2 in the pathogenesis in AMD. This has the following reasons:

1. The authors state: "It was previously reported that AKT2 is upregulated in dysmorphic macular RPE cells from AMD donor globes 13,14". Paper 13 shows for human retinas with AMD features (dysmorphic RPE?) increased LCN-2 expression, a transcription factor that drives inflammation; and this is what the paper is about. LCN-2 might be activated by AKT2 and NFkappaB but these are not the only pathways.

We apologize for the confusion. It has been previously reported that LCN-2 functions as a pro-inflammatory adipokine rather than a transcription factor (PMID: 25257511; 28026019; 31901947; and 35473441). We have previously reported that Akt2 expression is increased in AMD eyes, as well as in RPE cells from a mouse model exhibiting an AMD-like phenotype (Ghosh et al., 2017; J Pathol). We further observed that Akt2 upregulation subsequently triggers NFkB-dependent LCN-2 expression. These studies, highlighted in reference 13 (PMID: 28026019), provide strong support for the upregulation of AKT2 in dysmorphic RPE cells (PMID: 28026019 and 35473441).

Paper 14 reported a role of miRNA184 in RPE differentiation; a micro-RNA that targets AKT2 expression among other RNAs. The group showed in an eye from one donor increased ATK2 mRNA levels compared to one age-matched control. The authors concluded a dysfunctional RPE, but I do not know dysfunctional in which way. The authors did not show decreased levels of miRNA184. It is rather so that RPE cells are dedifferentiated in AMD. As only one donor is analyzed, we have clue whether this is a regular observation in AMD.

Reference 14 was used to acknowledge previous observations demonstrating AKT2 expression and AMD. We agree with the reviewer that the findings described in reference 14, from one AMD eye and one control eye, are intriguing but also insufficient to draw conclusions on AMD pathology. AKT2 plays a central role in many cellular functions and can be regulated by several intracellular signaling pathways, including miRNAs (as reported in reference 14). Whether these diverse pathways and/or microRNAs converge on AKT2 in RPE during AMD pathogenesis is of importance, but it is beyond the scope of the current study.

2. AKT2-KI mouse model shows dystrophic RPE cells with age that is more frequent than in the WT controls. Age-dependent changes indicate faster progression of a degenerative phenotype compared to WT animals. This is very interesting but an AMD-like phenotype by overactivity of AKT2 in mouse does not prove that AKT2 is causing AMD in humans. It can be that AKT2 mechanisms have secondary effects due to ongoing degeneration. Furthermore, AKT2 is a highly important kinase in

photoreceptors. Li et al. (2007 J Neurosci) showed that AKT2 acts in a non-redundant manner together with AKT1 in the protection of photoreceptors against light-damage. Thus, an impact of overactive AKT2 in photoreceptors should not be overseen when claiming that AKT2-KI is a RPE-based model for AMD.

The authors have previously shown (PMID: 36229454) that the *Akt2* KI mice have RPE-specific upregulation of AKT2 driven by the *Best1* promoter. In this manuscript the authors show that due to the upregulation of AKT2 in the RPE, a dry AMD-like phenotype develops with age in these mice (Figure 1 and Supplementary Figure 1). Nonetheless, we agree that AKT2 plays important roles in other cells that are likely critical for retinal function, as the reviewer very rightly points out (with reference; Li et al 2007). However, it is important to note that the *Akt2* KI mouse model utilized in our investigation exhibits RPE-specific upregulation of AKT2 (PMID: 36229454 and Supplementary Figure 1a), and multiple publications in journals of high repute have also previously shown that RPE cells can also drive AMD pathogenesis and as the disease progresses the retina/photoreceptors become affected (PMID: 29636475, PMID: 35921555, PMID: 28228282, PMID: 24468901, PMID: 28083894, PMID: 36835257, PMID: 35042858).

These findings suggest that the AMD-like phenotype observed in our aging *Akt2* KI mouse model results from RPE dysfunction rather than overactive AKT2 activity in photoreceptors.

3. The link for human AMD should be given by the usage of iPS-RPE cells from a donor carrying the risk allele CFH-Y402H. The authors claim this to be a “disease in the dish model” for AMD because AMD starts in the RPE. This argument is questionable because other observations show that AMD might also start in the photoreceptors or in the choroid.

We agree with the reviewer that neither the initial insult (aging, genetic, oxidative stress, inflammatory, hypoxia, etc.) nor its location are known for certain. However, we believe that there is mounting evidence implicating the RPE as the cell/tissue driving disease onset in early AMD. In a recent paper studying human AMD eyes, the investigators reported a global decrease in chromatin accessibility in the RPE in early/dry AMD eyes, while in advanced disease this was also present in the neurosensory retina (PMID: 29636475). This finding suggests that dysfunction in the RPE may precede photoreceptor injury in AMD. Several other publications also suggest that RPE dysfunction initiates disease progression (PMID: 35932581; 35921555; 28228282; 24468901; 28083894; 36835257; and 35042858).

The “disease in the dish model” for AMD developed by Dr. Kapil Bharti’s lab at the NEI has been helpful in improving our understanding of AMD pathogenesis (PMID: 36550275; 30651323; 34911940; 35880133; and 35522714) as well as for Stargardt’s disease (PMID: 36306781). We therefore used this model here to help corroborate our extensive *in vivo* and *in vitro* studies. Nonetheless, we agree with the reviewer that AMD is a complex disease involving many cells/tissues. We have therefore revised the text to emphasize the strengths – as well as the limitations – of the iPS-RPE cell model. We thank the reviewer for raising this important point.

That the iPS-RPE cell culture represents a model for AMD is supported by a paper (reference 20) that has not been replicated so far.

The iPSC-RPE cell culture model utilized in this study used an established protocol developed by the laboratory of Dr. Kapil Bharti (a coauthor on this manuscript). This protocol, validated by several studies from Dr. Bharti’s lab, as well as several other leading AMD research groups, has proven to be a robust RPE model for investigating AMD disease pathways and progression *in vitro* (PMID: 34911940; 33918210; 37126685; 30128491; 28913923; 33918210; and 36306781).

The paper that is cited here shows in one iPS-RPE cell line with CFH-Y402-H that somehow complement is dysregulated leading to increased levels of terminal complement complex (C5b-9) in

lysosomes which in turn leads to lysosomal dysfunction. The cited paper leaves many questions open and does not provide a valid link between CFH polymorphism and lysosomal dysfunction (C5b-9 detection was performed with anti-C9 antibodies; can be that it is C9 and not C5b-9 in the lysosomes).

We apologize for the confusion. The authors referenced the Cerniasuskas 2020 paper (reference 20) to acknowledge prior work utilizing CFH Y402H RPE cells for studying lysosomal function. While we agree that it remains unclear which specific antibody was utilized and whether it effectively targets the relevant complement factor (C9 or C5b-9), the experimental results presented in the publication suggest lysosomal dysfunction in CFH Y402H cells. Despite ambiguities in this study, it is considered significant in the AMD field and was therefore appropriately cited.

4. The authors in this study, did not try to at least recapitulate the data from reference 20 to support at least the relevance of iPS-RPE with CFH-Y402H so that the lysosomal dysfunction is comparable and a general feature in in this model. However, the authors provide data how AKT2 might lead to dysfunctional lysosomes and claim that they provide the link between CHF polymorphism and lysosomal dysfunction.

Similar to the observations in reference 20, previous studies have indicated the accumulation of complement factors (C3 and C5) in CFH Y402H RPE, as demonstrated in iPSC-derived RPE cells from CFH Y402H-carrying donors (PMID: 34911940). The goal of our study was to build upon (rather than reproduce) the findings from these studies. In our study, the authors present significant supporting evidence for lysosomal dysfunction in CFH Y402H cells, consistent with the findings of reference 20, including increased expression and activity of CTSD and CTSL (please see Figure 2a-c). Furthermore, the authors have shown in the revised manuscript elevated levels of master regulators associated with lysosomal function/biogenesis, namely p-TFEB and p-TFE3, in CFH Y402H cells, consistent with a decrease in the nuclear translocation of these transcription factors (as shown in Figure 2a), providing additional evidence of abnormal lysosomal biogenesis in these cells.

It remains unclear how CFH-Y402H activated AKT2. The authors state that polymorphic CFH accumulates in lysosomes as a mechanism. The focus onto CFH-Y402H is important because the polymorphism that is the most relevant difference between the used cell line models. There is no evidence for a mechanism how CFH (not C5b-9) lysosomal accumulation activated AKT2. Thus, there is a missing link between the mouse model and the human condition.

We apologize for the confusion. The authors are not suggesting that CFH accumulation in lysosomes results in activation of AKT2. We further assert that we do provide a link between the mouse model and the human condition. It is established that CFH polymorphism leads to the accumulation of both C3 and C5 in RPE cells (PMID: 33751148). The accumulation of C3 and C5, in turn, activates the alternate complement pathway around RPE cells. Activation of the complement pathway triggers several intracellular signaling cascades in the RPE, which lead to activation of multiple kinases (e.g., AKT, ERK, and AMPK) and transcription factors (e.g., NFκB) (PMID: 34911940, PMID: 34943047, PMID: 35056119), with particular emphasis on AKT2-dependent lysosomal dysfunction, as detailed in this manuscript. Activation of these pathways is believed to initiate an AMD-like phenotype both in mouse and *in vitro* models (PMID: 34911940, PMID: 30808757).

5. The authors neglect a large body of evidence about the role of CFH-Y402H in AMD. In the light of this evidence their own data lose significance. This body of evidence clearly demonstrates that the CFH-Y402H role in the AMD pathology lies in the extracellular control of the complement system. Among these observations are those that indicate increased levels of C5b-9 and other complement factors in the plasma/serum from AMD patients. Locally, accumulation of complement components in

the outer retina (Drusen, Bruch's membrane etc.) appeared characteristic (excellently reviewed by Clark and Bishop (2018 Semin. Immunopathol) or Toomey et al. (2018 ProgRetinEyeRes). The complement activity regulation under CFH-Y402H shows weak control of C3 convertase and thrombospondin activity. For that reason, the link between CFH-Y402H, lysosomal dysfunction and AKT2 must be stronger demonstrated.

We thank the reviewer for raising this important point. The reviewer highlights the importance of extracellular complement factors in the progression of AMD. We agree that this is likely a driving factor for the contribution of CFH-Y402H to AMD development. However, while extracellular complement exacerbates AMD phenotypes in mouse models and *in vitro* systems, it has also been reported previously that abnormal intracellular signaling resulting from risk alleles and the subsequent buildup of C3/C5 could contribute to early changes in RPE associated with AMD (PMID: 34911940). Interestingly, this intracellular mechanism is exacerbated by the presence of extracellular complement sources (PMIDs: 34911940 and 24702844). In response to the reviewer's suggestion, in the revised manuscript we have included additional experiments to investigate the influence of systemic complement factors on Akt2-dependent lysosomal function. These experiments (please see Figure 5f) involved culturing CFH Y402H cells with CCHS (complement competent human serum), which serves as a source for anaphylatoxins (i.e., activated complement), mimicking the age-induced increase in the alternate complement pathway observed in AMD eyes (PMID: 34911940). To further characterize the contribution of AKT2, the authors conducted additional experiments in which they used an AKT2 inhibitor treatment, on CFH (H/H) +CCHS cells (please see Figure 5f and Supplementary Figure 10), to ascertain lysosomal function and lipid accumulation (major phenotypic change in AMD models both *in vivo* and *in vitro* (PMID: 34911940, PMID: 30808757, PMID: 24468901). Results from these studies demonstrate that CCHS treatment leads to the accumulation of lipid molecules that is rescued upon AKT2 inhibitor treatment (please see Supplementary Figure 10). Additionally, treatment with either the AKT2 inhibitor or trehalose also rescues lysosomal function in these cells (please see Figure 5f). Furthermore, the authors have also included data showing rescue of lysosomal function upon trehalose or AKT2 inhibitor treatment in CFH (H/H) cells derived from AMD donors (Fig. 5g and h).

Support of the conclusions by the data

A couple of data weaken the conclusions:

1. The AKT2-KI phenotype: I am somewhat puzzled by the data. At month 12 the WT mice show a dense rhodopsin staining in the IS/OS. In the AKT2 mouse of the same age there apparently no rhodopsin present. The thickness of IS/OS seems to be normal. Thus, there is no photoreceptor degeneration but loss of rhodopsin. Therefore, AKT-2 overactivity has a severe impact onto photoreceptors. In the light of this observation, the AKT2 overactivity in the RPE loses its relevance.

The reviewer raises an important point. Maintaining RPE homeostasis is crucial for preserving photoreceptor health. The RPE actively interacts with photoreceptors to sustain visual function (PMID: 15987797; 33281830; and 21091424). The observed phenotype in the *Akt2* KI mouse (loss of rhodopsin) is therefore consistent with the critical role of a healthy RPE to promote photoreceptor health and function. Importantly, our *Akt2* KI mouse model limits AKT2 overexpression specifically to the RPE; there is no impact on expression of Akt2 in the retina (PMID: 36229454 and Supplementary Figure 1a). We therefore assert that the observed effects on photoreceptors in this RPE-specific *Akt2* KI model are secondary to abnormalities in the RPE. We apologize if this was not adequately explained in the text. We have revised the text to make this clear. To further substantiate photoreceptor degeneration, the revised manuscript also includes outer nuclear layer (ONL) thickness measurements from age-matched WT and *Akt2* KI retinal sections which demonstrate decrease in ONL thickness in aged *Akt2* KI retina (Supplementary Figure 1g).

Indeed, AKT2 is highly active in photoreceptors (Li et al. 2007 J Neurosci).

Thank you for raising this point. Our *Akt2* KI mouse model is an RPE-specific KI model in which we specifically overexpressed *Akt2* in RPE cells (not in the neurosensory retina), as demonstrated previously (PMID: 36229454) and in Supplementary Figure 1a. Consequently, the observed effect on photoreceptors in this mouse model with age are secondary to the abnormalities in the RPE. We apologize for the confusion.

2. Why there are data from the mouse in a mix of 12 months and 15 months old animals? Is there a bias for selected results?

In this study, we have thoroughly examined the phenotype of *Akt2* KI mice (please see Figure 1 and Supplementary Figure 1). The AMD-like phenotype emerges at 10 months of age in our *Akt2* KI mouse model and was unchanged through 15 months of age. Consequently, we conducted experiments on aged mice ranging from 10 months (the earliest time point for when the phenotype was observed) to 15 months (aged mice with the same AMD phenotype). We included multiple replicates across various ages for each experiment, as outlined in the manuscript and the results were consistent within this time frame. Importantly, for all experiments, all animals were age-matched preventing bias from selecting animals at different ages. For accuracy we reported the ages of the mice used in the specific representative experiment used in the figure.

3. Is there a way to show the complete western-blot and not only single bands?

The raw western blot files will be uploaded and available to readers as per Nature publication policies. Including them in the main figures would cause space constraints.

4. Conditional knockout: It seems that the Cre-Lox system is used. Cre expression in the RPE has been found to be toxic for the RPE; applies especially for the *Cryba1* model.

In this study we used the Cre-LoxP system to generate the RPE-specific *Akt2* and *Cryba1* cKO mice using the *Best1-Cre* (PMID: 36229454 and 24468901). We have previously published multiple articles validating the RPE-specific *Cryba1* cKO using the Cre-LoxP system as a mouse model with AMD-like phenotype (PMIDs: 30098172; 35473441; and 37848361), which has been acknowledged as a valuable mouse model for dry AMD in review articles on models of AMD (PMID: 35835183 and 33993621). Of note, the global knockout of the *Cryba1* gene, in which the Cre-LoxP system is not used, also shows an AMD-like phenotype with age (PMID: 28083894), suggesting that the phenotype observed in the *Cryba1* cKO mice model is not likely to be due to Cre toxicity. To rule out a contribution of Cre toxicity in the RPE-specific *Akt2* KI mouse model, we have included multiple controls, including studies in *Best1-Cre* mice, in which Cre is expressed in the RPE, but in which we do not see a similar phenotype as the *Best1-Cre/Akt2* KI mouse. We are therefore confident that the phenotype we report here is due to *Akt2* overexpression rather than Cre toxicity.

5. Cathepsin D/L activities in human AMD retina: this is an important piece of the data. However, it is to mention that three controls and three AMD retinas were analyzed. The data show that the activities of cathepsin D and L were decreased in the AMD probes. This can have other reasons than reflecting a direct disease mechanism. RPE cells in geographic atrophy degenerate and display a reduced metabolism with decreased protein expression and phosphorylation activities. Thus, cathepsin activity reduction might be effect of the degenerative processes and not cause.

The reviewer raises an important point. We agree that extensive degeneration occurs in geographic atrophy, thus studying disease progression with such samples may be misleading. To avoid this, the human cadaver samples utilized in this study were obtained from donors with early

AMD. Grading was conducted using the Minnesota grading system (MGS), as previously described (PMID: 28026019), to categorize disease stages as MGS1 (no disease), MGS2 (early/dry AMD), MGS3 (geographic atrophy), and MGS4 (neovascular AMD). We also agree that the reduction in cathepsin activity can be an effect of the degenerative processes and not the cause. However, in young *Akt2* KI mice (at an age where no retinal degenerative changes are observed) and in CFH Y402H RPE cells (derived from donors carrying the risk allele, not from those with the disease), we observed diminished levels of CTSD and CTSL (please see Figure 1d). These findings suggest that deregulated nuclear activity of these transcription factors leads to a decline in CTSD and CTSL levels, and the decline in the levels of these lysosomal factors is not solely because of the degeneration.

Why did the authors not measure AKT2?

Agreed. In response to the reviewer's suggestion, we have included the levels of p-AKT2 in human AMD donor samples in the revised manuscript. The results indicate a significant increase in the pAKT2/AKT2 levels in the AMD donor RPE compared to age-matched controls (please see Figure 2d).

6.IP experiments: When IP was performed with an anti-GFP antibody AKT2 can be precipitated when using the GFP-AKT2 plasmid transfection but becomes weaker in the GFP-AKT2/SIRT5-HA double transfection. Why? Or is this just an observation in this blot?

Thank you for raising this point. To clarify, the decreased levels of GFP in double plasmid transfected (GFP-AKT2/SIRT5-HA) cells (New Supplementary 9b) was not significant across all the 4 biological replicates used for this experiment and therefore not reported as a noticeable change in the original submission.

7.CFH KO mouse: this model needs to be taken with care. Due to the lack of CFH the mice are completely lacking C3. It is hard to differentiate between the lack of CFH and C3.

The authors would like to state that this study utilized only *Cfh* null RPE cells rather than CFH KO mice.

8.In their previous paper, the authors showed that AKT2 KO in the mouse leads to compensatory upregulation of AKT1 that in turn participates the pathology of diabetic retinopathy. Is AKT1 downregulated in AKT2 KI mice? Can it be that the pathology lies on a AKT1 -dependent pathway?

The authors have previously demonstrated that in RPE cells from *Akt2* KI mice, there is no alteration in the levels of AKT1. This suggests that compensatory mechanisms between the two AKT isoforms are primarily observed in cases where there is a loss or knockdown of one isoform. This evidence is supported by Supplementary Figure 5d and e of PMID: 36229454. Hence, the pathology observed in *Akt2* KI is not mediated through an AKT1-dependent pathway.

9.iPS-RPE cells: Among the cell lines this model is widely accepted as being a very relevant model. However, also these cells bear strong quality impairments. The paper lacks a validation of the cell line, that is also several times passaged. First, the authors used only one cell line from one donor. It is known that even different cell lines from the same donor display different expression activities of RPE markers (see as an example Marmorstein et al. 2018 Sci Rep).

In the manuscript, we have clearly shown the utilization of cells derived from three individual donors to generate a biological replicate of 3 (n=3). This information has been appropriately included in both the methods section and figure legends for clarity and transparency. Moreover, in response to

the reviewer's suggestion, we have improved the validation of these cell lines by including baseline results for TER and RPE marker (Ezrin, PMEL17) expression (Supplementary Figure 4). We have also included genotype information for each cell lines used in this study (Supplementary Table-1 and 3). These additional data provides further validation and robustness to our experimental approach. We believe these additions strengthen the rigor and reliability of our findings.

We cannot exclude that the differences between the CHF-Y402H cell line and the CFH-WT cell line are due to variances in gene expression of lysosomal pathways or AKT2 signaling.

In our revised manuscript, we present data demonstrating that the variances in the expression of lysosomal mediators, such as cathepsin D and L, in CFH Y402H cells compared to WT cells are primarily attributed to abnormal TFEB signaling, a transcriptional regulator of lysosomal biogenesis/function. Furthermore, to substantiate our claim that abnormal expression of these lysosomal mediators is linked to upregulated AKT2 signaling, we have included new experimental findings. Specifically, inhibition of AKT2 has been shown to rescue the abnormal expression of lysosomal hydrolases like cathepsin D and LAMP1 in CFH Y402H cells (see Figure 5f).

iPS-RPE react very sensitive on passaging, another inducer of variability. The investigation of one cell line from one donor and one from a control donor is not enough for generalizations.

As stated above, cells were obtained from three different donors to create three biological replicates (n=3). The quality of iPS-derived RPE cells was previously confirmed by several publications, as referenced by the following publications: PMID: 36550275, PMID: 30651323, PMID: 34911940, PMID: 35880133, and PMID: 35522714.

And a very important question, when it comes to lysosomes: are the iPS-RPE used in this study pigmented?

In this study, we used only pigmented RPE cells, a characteristic indicative of their differentiation. The accumulation of pigment serves as a hallmark feature distinguishing differentiated RPE cells. Notably, these RPE cells exhibited functional activity, mirroring all key features of human RPE, as corroborated by previous studies (PMID: 30651323, PMID: 27400791). Additionally, the authors have included the expression of PMEL17 (known pigmented melanosome marker in RPE) in iPSC-derived RPE cells (Supplementary Figure 4), to further prove the presence of pigmentation.

6. AKT2 overactivity in mouse: The authors present a large body of data that precisely describe the chain of events that lead to impaired lysosomal function, and in turn to increased secreted autophagy by RPE cells. Indeed, Figure 1 shows kind of basal lamina deposits in the AKT2-KI. However, how can the authors exclude basal laminar deposits in the 15 months old WT? Furthermore, At the end of the results part the authors claim that Trehalose treatment abolishes the basal laminar deposits. This is not shown.

A 15-month-old WT mouse, or mice of any age, are unlikely to exhibit BlamD accumulation. This specific change is typically observed in mouse models with an AMD-like phenotype, as previously indicated by our laboratory and others (PMID: 28083894, PMID: 30808757, PMID: 35042858). In Figure 6i, the results clearly demonstrate that trehalose treatment of *Cryba1* cKO mice rescues the ONL changes and sub RPE deposits compared to animals treated with the vehicle, consistent with our previous findings (PMID: 31552301). Furthermore, for improved readability, the authors have included quantification of this change in the *Akt2* KI mice as well, in the revised manuscript (Figure 6h),

7. Most of the Trehalose intervention studies were performed with the *Cryba1* mouse model.

Especially the most relevant data on retinal structure. However, this does not proof the AKT2 hypothesis. Why were Trehalose experiments not done with AKT2-KI?

The authors would like to clarify that previous findings have demonstrated an increase in Akt2 expression within the RPE of the *Cryba1* cKO mouse model, and inhibiting Akt2 has been shown to ameliorate early RPE changes in this model (PMID: 31552301, PMID: 28026019). Therefore, the trehalose treatment to *Cryba1* cKO mice and the AKT2 hypothesis is justified. Additionally, evidence of trehalose-mediated rescue in CLEAR genes, lysosomal function and autophagy mediators has been depicted in RPE cells from *Akt2* KI mice following *in vivo* treatment in the original submission (Figure 6d-g). Additionally, quantification of ONL nuclei in hematoxylin-eosin stained retinal sections in both *Akt2* KI and *Cryba1* cKO mouse models with and without trehalose treatment has been included in the revised manuscript, showing noticeable rescue from retinal degenerative changes upon trehalose treatment (Figure 6h,i).

12. There is a spidergram in the last figure that shows retinal degeneration in the *Cryba1* model. This not shown for AKT2. However, given the retinal sagittal sections, I can not recapitulate the IS/OS ratios. Why not counting the number of rows of nuclei in the ONL?

The authors would like to thank the reviewer for this suggestion. The authors have incorporated a quantification in the revised manuscript for *Cryba1* cKO and *Akt2* KI mice +/- trehalose treatment, based on the number of ONL nuclei. This result demonstrates a significant rescue effect compared to untreated animals (Figure 6h, i).

Gender

Epidemiology shows differences between male and females in the incidence and risk for AMD.

Gender differences are of importance.

Human tissue: No gender data of the donors used for analysis of human tissue; only three with AMD and three age-matched controls.

iPS-RPE: no data whether these are male or female cells.

Mouse experiments: No indications about the gender of mice used in the study.

Gender-specific bias in AMD prevalence is noted in certain studies, but research with large sample sizes and diverse population cohorts fails to demonstrate any such bias (PMID: 25104651, PMID: 29346644). This study utilized both female and male mice and donors, randomized into experimental groups as outlined in the revised manuscript's Methods section. No sex-dependent trends were observed in this study.

Reviewer #3 (Remarks to the Author):

This is an interesting study describing the potential role of AK2/SIRT5/TFEB pathway in lysosomal dysfunction in AMD. The strength of the study is the use of different models, namely the Akt2 knock-in mouse, AMD patient Y402H RPE cells, AMD donor eyes and ARPE19 cells to demonstrate accumulation of AK2, which impairs TFEB/TFE3 lysosomal function via inhibition of PGC-1 α and SIRT5. While the work has been carefully conducted, there are details missing in both text and figure legends, which should be added at the revision stage. Further, the data has not been widely discussed in the context of other publications and thus there is scope to balance their findings against previous work that have not only shown lysosomal dysfunction in AMD RPE cells, but also provided mechanistic insights. Below there are some questions/comments that will strengthen the manuscript:

1. How many iPSC-RPE and isogenic controls were used for the study? Apart from the Y402H polymorphism, were there any genotype changes in other complement genes (e.g. C3, C5)? If yes,

were there any changes in AK2 expression between iPSC-RPE derived from AMD patients with only Y402H and Y402H+ other complement risk alleles?

The authors would like to thank the reviewer for the helpful suggestion. The authors have included detailed Supplementary Tables (1 and 3) to summarize the cell lines used with their respective genotypes in the revised manuscript. The figure legends have been modified to indicate the 'n' for each experiments utilizing these cell lines. Importantly, there were no genotype alterations observed in other complement genes among the cell lines employed for this investigation.

2. The authors show changes in pTFE-3 and pTFEB expression in AMD donors: this work needs to be validated in the Y402H-RPE cells and in the Akt2 KI RPE cells.

Following the reviewer's very helpful recommendation, the authors have incorporated measurements of p-TFE3 and p-TFEB levels in both CFH Y402H and *Akt2* KI RPE cells into the revised manuscript. The results revealed a significant elevation in the protein levels of both p-TFE3 and p-TFEB in Y402H and *Akt2* KI RPE cells, as illustrated in Figure 2a and Supplementary Figure 2a.

3. Since TFEB is a master regulator of lysosomal biogenesis and the authors are describing lysosomal dysfunction, one would have expected changed in lysosome number and/or volume/surface area. None of these parameters have been described. These data can be easily obtained from the existing TEM data.

The authors have incorporated quantification for lysosomal area using an AI-based technique on the TEM micrographs in the revised manuscript, following the suggestion of the reviewer. This analysis revealed an increase in the lysosome area normalized to the total tissue area in *Akt2* KI mice compared to WT mice across ages ranging from 3 to 15 months, as depicted in Supplementary Figure 2b.

4. In figures 5 and 6 changes in expression of CTSL and CTSD have been described in response to various treatment, however, to reach the conclusion that lysosomal function has been rescued, enzyme activities need to be carried out to exclude the potential scenario of these proteins being secreted in the media.

In response to the reviewer's suggestion, the revised manuscript now includes the activities of CTSD and CTSL in CFH (H/H) iPSC-derived RPE cells and *Akt2* KI RPE cells under various treatment conditions. These findings are presented in Figure 5d, e and Figure 6f, g respectively.

5. Figure 1c: higher magnification images are needed to substantiate the presence of Blam deposits.

The reviewer has raised a valid point. However, the authors wish to clarify that higher magnification images are not required to confirm the presence of BlamD. The TEM images presented in Figure 1c, featuring a scale bar of 600 nm, suffice for this purpose. This is because BlamD deposits in mouse models of AMD have been demonstrated to fall within a maximum height range of approximately 4 μm , detectable at TEM magnifications of 1 μm (PMID: 25991857). Additionally, in the original submission, the presence of BlamD in the *Akt2* KI mice was further confirmed through Perilipin2 staining (Supplementary Figure 1f), as previously described (PMID: 35042858).

6. Seahorse assays have only been performed in ARPE19 cells overexpression Akt2 or Ak2+Sirt5. To validate these findings, it would be necessary to repeat the seahorse assays in the Y402H and isogenic control derived RPE cells.

Agreed. Generating and maintaining iPSC cells poses significant challenges. Hence, we collaborated with Dr. Bharti's lab at NEI, who is a co-author on the manuscript, to address this issue. Additionally, we did not conduct a Seahorse assay on the CFH Y402H cells (characterized by high AKT2 and low SIRT5 levels; as depicted in Figure 1g and 3h). This decision was based on the prior demonstration of comparable data in RPE cells obtained from AMD patients carrying the Y402H risk allele (PMID: 33918210).

7. Flores-Bellver described recently increased exocytosis in RPE cells treated with AMD stressors: the authors have made no attempt to discuss their findings in the context of these published data.

The authors have incorporated the findings from the Flores-Bellver manuscript into the Discussion section of the revised manuscript.

REVIEWER COMMENTS

Reviewer #1 (Remarks to the Author):

The authors did a very good job in addressing the large number of comments that were raised. However, I would still like to draw the author's attention to somewhat problematic responses to point 7 and 14.

7) I understand and accept that fractionation of the small amount of tissue is problematic. However, the total expression of a protein (Fig. 2D) does not reflect its localization. Therefore, the results of translocation remain unconvincing. Since this point is not essential for the paper, I suggest addressing this minor point in the text.

14) The authors should note that the phosphorylation of AKT is mediated by PDK1 and mTORC2, not PI3K. The phosphorylation of S474 by itself is not sufficient to induce full AKT2 activation, which also requires the phosphorylation of T308. PDK1 and probably mTORC2 are solely membranal, and therefore, the mechanism of AKT2 phosphorylation is not clear. This point is important and should be addressed/discussed prior to publication.

Reviewer #2 (Remarks to the Author):

The authors did well address the points of criticism by the reviewers. Additional experiments have been performed. I have some minor points.

Reviewer #1

15: If Akt1 and/or Akt3 cannot be excluded as both the authors and reviewer concur then I suggest to address this in the text: "A role of Akt1 and Akt3 cannot be excluded but our data justify a role of Akt2".

Reviewer #2

1. Paper 14: When the authors agree that the conclusions in Paper 14 are not relevant for AMD then they should either comment the weakness or leave the paper out. Furthermore, when this citation was meant in the context that AKT2 regulate miRNA then other papers are more appropriate.

3. The only RPE hypothesis: As in all conclusions I would like to ask the authors to be more careful. No one doubts that the RPE is involved in the chain of events leading to AMD. But that the disease origins in the RPE is not clear. A couple of publications highlight age-dependent loss of rods and cones (e.g. DOI: 10.1038/eye.2001.140) but over the same time span no loss of RPE cells in the macula (e.g. DOI: 10.1167/iovs.14-14802; PMID: 12356840). In contrast, over this time span the choroid becomes thinner and Bruch's membrane thicker (e.g. PMID: 11133878; DOI: 10.1097/IAE.0000000000001347).

- PMID: 33751148): Here the authors claim that "it is established that CFH polymorphisms lead to accumulation of C3 and C5 in the RPE". The paper cited here is a review article. Nowhere in this article is an accumulation of C3/5 in the RPE mentioned. Furthermore, C3 or C5 presence leads activate the alternative complement activation pathway is not correct. There is no mechanism known how C3 activates the alternative pathway. It is better to understand that CFH polymorphisms associate with a higher risk in AMD and does not cause AMD in a sense of a monogenetic disease. The polymorphism leads to a less effective inhibitory control of the alternative pathway. So what might be possible is that active complement components such as C3a or C5a activate AKT in the RPE.

-

Reviewer #3 (Remarks to the Author):

The authors have answered all my questions and comments.

REVIEWER COMMENTS

Reviewer #1 (Remarks to the Author):

The authors did a very good job in addressing the large number of comments that were raised. However, I would still like to draw the author's attention to somewhat problematic responses to point 7 and 14.

The authors would like to thank the reviewer for acknowledging that we have addressed the large number of comments that the reviewers raised. Our responses to points 7 and 14 are detailed below.

7) I understand and accept that fractionation of the small amount of tissue is problematic. However, the total expression of a protein (Fig. 2D) does not reflect its localization. Therefore, the results of translocation remain unconvincing. Since this point is not essential for the paper, I suggest addressing this minor point in the text.

Agreed. The authors have acknowledged in the revised manuscript that phosphorylation does not reflect protein localization. However, we have also explained why phosphorylation of these transcription factors could be attributed to reduced nuclear activity of TFEB/E3 (please see lines 211-214).

14) The authors should note that the phosphorylation of AKT is mediated by PDK1 and mTORC2, not PI3K. The phosphorylation of S474 by itself is not sufficient to induce full AKT2 activation, which also requires the phosphorylation of T308. PDK1 and probably mTORC2 are solely membranous, and therefore, the mechanism of AKT2 phosphorylation is not clear. This point is important and should be addressed/discussed prior to publication.

Agreed. The authors have explained the importance of the two phosphorylation sites of AKT2 (S474 and T309; since S473 and T308 is for AKT1) in the discussion of the revised manuscript (please see lines 407-413).

Reviewer #2 (Remarks to the Author):

The authors did well address the points of criticism by the reviewers. Additional experiments have been performed. I have some minor points.

The authors would like to thank the reviewer for acknowledging that we have addressed the queries raised by the reviewers. The responses to the minor points are detailed below.

Reviewer #1

15: If Akt1 and/or Akt3 cannot be excluded as both the authors and reviewer concur then I suggest to address this in the text: "A role of Akt1 and Akt3 cannot be excluded but our data justify a role of Akt2".

Agreed. The authors have addressed the potential role of other AKT isoforms in the introduction of the revised manuscript (please see lines 103-105).

Reviewer #2

1. Paper 14: When the authors agree that the conclusions in Paper 14 are not relevant for AMD then they should either comment the weakness or leave the paper out. Furthermore, when this citation was meant in the context that AKT2 regulate miRNA then other papers are more appropriate.

The authors agree with the reviewer on this point. However, we would like to clarify purpose for citing paper #14 here which was to show that previous publications have also reported upregulation of AKT2 in human AMD; we did not intend to suggest that AKT2 has a role in miRNA regulation or vice-versa in the context of AMD pathogenesis.

3. The only RPE hypothesis: As in all conclusions I would like to ask the authors to be more careful. No one doubts that the RPE is involved in the chain of events leading to AMD. But that the disease origins in the RPE is not clear. A couple of publications highlight age-dependent loss of rods and cones (e.g. DOI: 10.1038/eye.2001.140) but over the same time span no loss of RPE cells in the macula (e.g. DOI: 10.1167/iovs.14-14802; PMID: 12356840). In contrast, over this time span the choroid becomes thinner and Bruch's membrane thicker (e.g. PMID: 11133878; DOI: 10.1097/IAE.0000000000001347). - PMID: 33751148): Here the authors claim that "it is established that CFH polymorphisms lead to accumulation of C3 and C5 in the RPE". The paper cited here is a review article. Nowhere in this article is an accumulation of C3/5 in the RPE mentioned. Furthermore, C3 or C5 presence leads activate the alternative complement activation pathway is not correct. There is no mechanism known how C3 activates the alternative pathway. It is better to understand that CFH polymorphisms associate with a higher risk in AMD and does not cause AMD in a sense of a monogenetic disease. The polymorphism leads to a less effective inhibitory control of the alternative pathway. So what might be possible is that active complement components such as C3a or C5a activate AKT in the RPE.

Agreed. In the revised manuscript, the authors have discussed the only RPE hypothesis and the implication of the *CFH* risk allele in context of AMD, as suggested by the reviewer. As per the reviewer's suggestion, the discussion of the revised manuscript has also been edited to elaborate on previous findings which have shown the importance of other cell types, like the photoreceptors and the choriocapillaris, in AMD pathogenesis (please see lines 383-390). We have also emphasized the importance of the *CFH* polymorphism as the highest risk allele associated with AMD (please see lines 432-437).

Reviewer #3 (Remarks to the Author):

The authors have answered all my questions and comments.

The authors would like to thank the reviewer for acknowledging that we have addressed all the questions.

REVIEWERS' COMMENTS

Reviewer #1 (Remarks to the Author):

The authors nicely addressed all my remaining comments.

Reviewer #2 (Remarks to the Author):

The reviewer thanks the authors for their further successful improvement of the manuscript.

The authors would like to thank the reviewers.